# Body size-dependent energy storage causes Kleiber's law scaling of the metabolic rate in planarians

Albert Thommen[1,2†], Steffen Werner[2,3†], Olga Frank[1†], Jenny Philipp[4], Oskar Knittelfelder[1], Yihui Quek[2,5], Karim Fahmy[4], Andrej Shevchenko[1], Benjamin M Friedrich[2,6], Frank Jülicher[2*], Jochen C Rink[1*]

[1]Max Planck Institute of Molecular Cell Biology and Genetics, Dresden, Germany; [2]Max Planck Institute for the Physics of Complex Systems, Dresden, Germany; [3]FOM Institute AMOLF, Amsterdam, The Netherlands; [4]Helmholtz-Zentrum Dresden-Rossendorf, Institute of Resource Ecology, Dresden, Germany; [5]Massachusetts Institute of Technology, Cambridge, United States; [6]Center for Advancing Electronics Dresden, Technische Universität Dresden, Dresden, Germany

**Abstract** Kleiber's law, or the 3/4 -power law scaling of the metabolic rate with body mass, is considered one of the few quantitative laws in biology, yet its physiological basis remains unknown. Here, we report Kleiber's law scaling in the planarian *Schmidtea mediterranea*. Its reversible and life history-independent changes in adult body mass over 3 orders of magnitude reveal that Kleiber's law does not emerge from the size-dependent decrease in cellular metabolic rate, but from a size-dependent increase in mass per cell. Through a combination of experiment and theoretical analysis of the organismal energy balance, we further show that the mass allometry is caused by body size dependent energy storage. Our results reveal the physiological origins of Kleiber's law in planarians and have general implications for understanding a fundamental scaling law in biology.
DOI: https://doi.org/10.7554/eLife.38187.001

*For correspondence:
julicher@pks.mpg.de (FJü);
rink@mpi-cbg.de (JCR)

†These authors contributed equally to this work

## Introduction

Body size varies strikingly across animal phylogeny. From small crustaceans weighing a few ng to blue whales weighing in excess of 140 000 kg, body mass variations span more than 16 orders of magnitude (*Makarieva et al., 2008*; *Sears and Calambokidis, 2002*). In spite of such tremendous variation in scale and physiology, the organismal metabolic rate (*P*; defined as the heat produced by the organism per unit time measured in Watts, which is related to the rate of oxygen consumption (*McDonald, 2002*)) nevertheless follows a general scaling relationship with body mass (*M*). As originally described by Kleiber in 1932 (*Kleiber, 1932*), *P* can be expressed by a power-law of the form $P = aM^b$, with *b* being the scaling exponent and a proportionality constant *a*. Although reported values of *b* vary somewhat between studies or specific animal species, a value of $b \approx 3/4$ is typically observed (*Banavar et al., 2014*; *Blaxter, 1989*; *Brody, 1945*; *Calder, 1984*; *Hemmingsen, 1960*; *Kleiber, 1961*; *Peters, 1983*; *Schmidt-Nielsen, 1984*; *West and Brown, 2005*; *Whitfield, 2006*) and this allometric relation between mass and metabolic rate is consequently referred to as the 'three-quarter' or 'Kleiber's law'. This implies that the specific metabolic rate (*P/M*) decreases as body mass increases, which is commonly interpreted as reflecting a size-dependent decrease of cellular metabolic rates. Surprisingly, despite being known since more than 80 years and termed one of

the few quantitative laws in biology (*West, 1999*), the physiological basis of Kleiber's law remains under intense debate.

The fact that all animals, irrespective of physiology, habitat or life style, obey Kleiber's law suggests a fundamental constraint in animal metabolism (*West and Brown, 2005*). Many hypotheses have been proposed that suggest a variety of origins of Kleiber's law. A major class of hypotheses are based on internal physical constraints (*Glazier, 2005*), for example space-filling fractal transportation networks (*West et al., 1997*) or size-dependent limitation of resource transport across external and internal body surfaces (*Davison, 1955*; *Maino et al., 2014*; *McMahon, 1973*). Another class of hypotheses concerns external ecological constraints, for example the optimization of body size for maximising reproductive fitness (*Koziowski and Weiner, 1997*). However, the experimental validation of the different hypotheses has proven difficult. Inter-species comparisons suffer from the difficulty of obtaining quantitative measurements in non-model organisms and from the often limited utility of comparisons between physiologically and genetically very distinct animals. Intra-species comparisons, that is comparisons between differently sized members of the same species, are often hampered by a limited size range and life history changes that profoundly affect metabolism (e.g., developmental transitions or aging). As a result, all hypotheses regarding the origins of Kleiber's law remain controversial also for the lack of a suitable model system.

Flatworm laboratory models offer interesting opportunities in this respect. Although usually studied for their regenerative abilities and pluripotent adult stem cells (*Reddien and Sánchez Alvarado, 2004*; *Rink, 2013*; *Saló and Agata, 2012*), the model species *S. mediterranea* and other planarians display tremendous changes in body size. They grow when fed and literally shrink (termed 'degrowth' in the field) when starving (*Baguñà et al., 1990*; *Oviedo et al., 2003*), which in *S. mediterranea* amounts to fully reversible body length fluctuations between ~0.5 mm and ~20 mm. Such a >40 fold range of body length in a laboratory model provides ideal preconditions for measuring the size-dependence of physiological processes. Moreover, the commonly studied asexual strain of *S. mediterranea* and other asexual planarians do not seem to age, thus rendering their reversible size changes independent of organismal aging (*Glazier, 2005*). Previous studies of metabolic rate scaling in planarians suggest a size-dependence of $O_2$-consumption (*Allen, 1919*; *Daly and Matthews, 1982*; *Hyman, 1919*; *Osuma et al., 2018*; *Whitney, 1942*), but the size dependence of *P* has so far not been systematically quantified.

We here report that metabolic rate scaling in *S. mediterranea* indeed follows Kleiber's law and we apply a combination of experiments and theory to understand its physiological basis. Our analysis of the organismal energy balance reveals that the size-dependent decrease in the specific metabolic rate does not reflect a decrease in the metabolic rate per cell, but instead an increase in the average mass per cell. Further, we demonstrate that the cell mass allometry reflects a size-dependent increase in lipid and glycogen stores. Our results therefore demonstrate that size-dependent energy storage causes Kleiber's law scaling in planarians.

## Results

### Planarians display Kleiber's law scaling of the metabolic rate

Kleiber's law describes the scaling of metabolic rate with the mass of animals. In order to test whether the tremendous body size fluctuations of *S. mediterranea* (*Figure 1A*) follow Kleiber's law, we needed to devise methods to accurately quantify the mass and metabolic rate of planarians.

To measure mass, we quantified both the dry and wet mass of individual planarians. Though dry mass measurements avoid the challenging removal of residual water from the mucus-coated animals, they are lethal and can therefore only be carried out once. As shown in *Figure 1B*, the wet and dry mass of *S. mediterranea* vary over > 3 orders of magnitude. Moreover, the near-constant ratio between wet and dry mass (~5; implying 80% water content) indicates minimal variations of the water content and thus facile interconversion of the two mass measurements.

In order to quantify the metabolic rate, we used microcalorimetry. Microcalorimetry measures the integrated heat generated by all metabolic processes inside the animal and therefore provides a pathway-independent measure of total metabolic activity (*Kemp and Guan, 1997*). The size-dependence of the metabolic rate was measured by enclosing cohorts of size-matched and two to three weeks starved animals in vials and measuring their heat emission over a period of > 24 h (*Figure 1—*

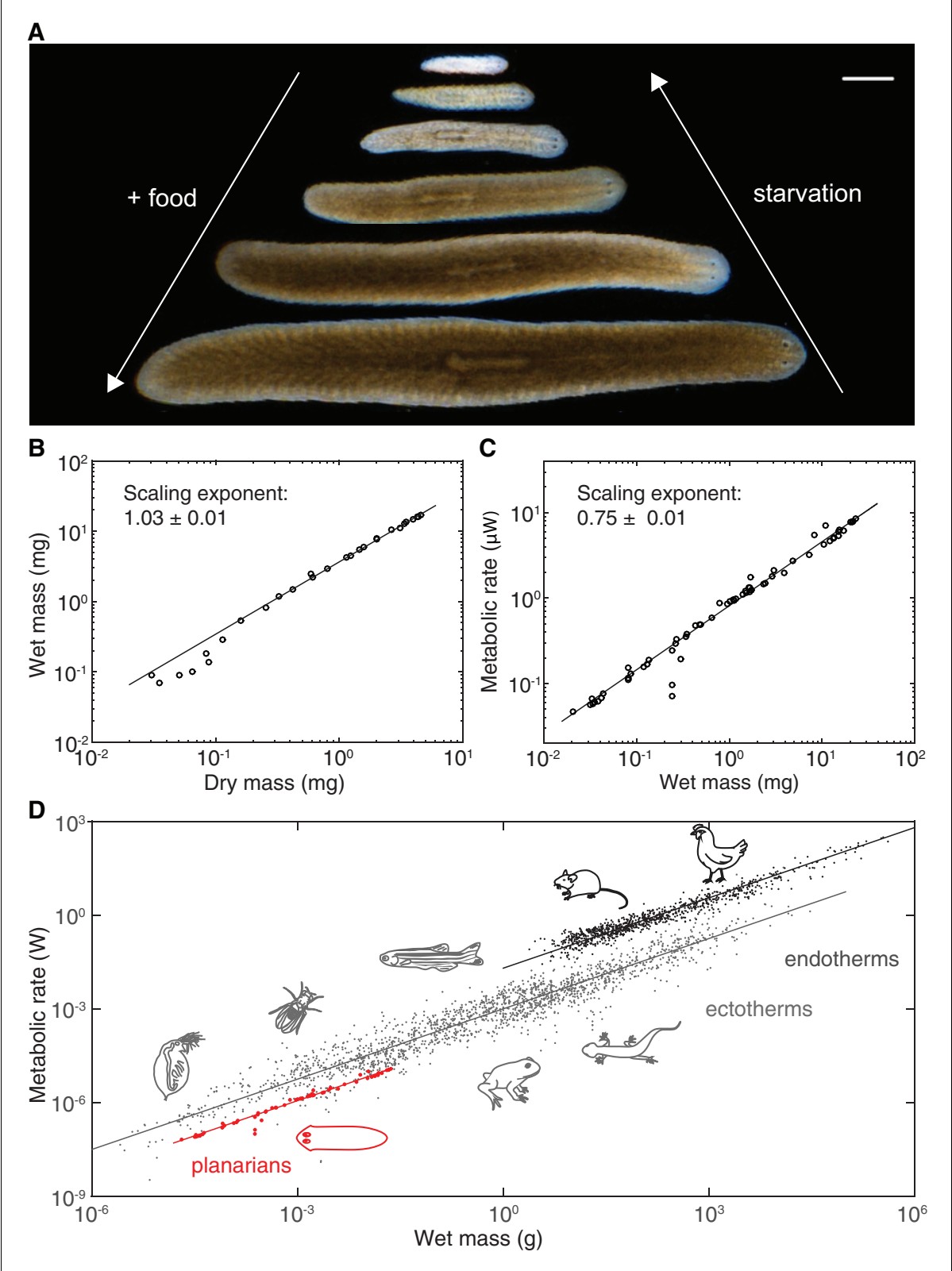

**Figure 1.** Kleiber's law scaling during *S.mediterranea* body size changes. (**A**) Feeding (growth) and starvation (degrowth) dependent body size changes of *Schmidtea mediterranea*. Scale bar, 1 mm. (**B**) Wet versus dry mass scaling with body size. The scaling exponent ± standard error was derived from a linear fit for wet mass > 0.5 mg and represents the exponent *b* of the power law $y = ax^b$. See *Figure 1—source data 1* for numerical data. (**C**) Metabolic rate versus wet mass scaling by microcalorimetry. The metabolic rate was determined by a horizontal line fitted to the stabilised post-

*Figure 1 continued on next page*

*Figure 1 continued*

equilibration heat flow trace (*Figure 1—figure supplement 1*) and the post-experimental dry mass determination of all animals in the vial was re-converted into wet mass by the scaling relation from (**B**). Each data point represents a vial average of a size-matched cohort. The scaling exponent ± standard error was derived from a linear fit and represents the exponent *b* of the power law $y = ax^b$. (**D**) Metabolic rate versus wet mass scaling in planarians from (**C**) (red) in comparison with published interspecies comparisons (*Makarieva et al., 2008*) amongst ectotherms (grey) or endotherms (black). Dots correspond to individual measurements; black and blue solid lines trace the 3/4 scaling exponent; red line, linear fit to the planarian data. By convention (*Makarieva et al., 2008*), measurements from homeotherms obtained at different temperatures were converted to 37 °C, measurements from poikilotherms and our planarian measurements to 25 °C, using the following factor: $2^{(25 °C - 20 °C)/10 °C} = 2^{0.5}$ (20 °C: planarian data acquisition temperature).

DOI: https://doi.org/10.7554/eLife.38187.002

The following source data and figure supplements are available for figure 1:

**Source data 1.** Numerical data wet mass vs. dry mass measurements.
DOI: https://doi.org/10.7554/eLife.38187.005

**Figure supplement 1.** Measurement of metabolic rate.
DOI: https://doi.org/10.7554/eLife.38187.003

**Figure supplement 1—source data 1.** Raw data metabolic rate measurements.
DOI: https://doi.org/10.7554/eLife.38187.004

*figure supplement 1*). Animal numbers per vial varied between 2 (= largest size cohort) and 130 (= smallest size cohort) in order to yield measurements with comparable signal-to-noise ratios. Since animals were not immobilized, our measurements effectively reflect the routine metabolic rate that is generally used for aquatic animals (*Dall, 1986*). As shown in *Figure 1C*, the metabolic rate measurements increase with mass over nearly 3 orders of magnitude (from ~0.02 to 10 μW). The data points can be fit with a single power law that accurately describes the size-dependence of the metabolic rate across the entire size range. Intriguingly, the value of the scaling exponent is 0.75 ± 0.01 and thus identical with the ~0.75 exponent associated with Kleiber's law in inter-species comparisons. Consequently, the slope of the planarian data points (red) exactly parallels the characteristic slope of extensive published data sets of specific metabolic rate measurements (*Makarieva et al., 2008*) (*Figure 1D*). While the offsets between endo- and ectotherm traces might reflect different temperature regimes as previously noted (*Hemmingsen, 1960*; *Makarieva et al., 2008*), the common slopes stresses the universal nature of the 3/4 law exponent across animal phylogeny. The fact that the same power law exponent is associated with the entire growth/degrowth-dependent body size interval of a planarian suggests that the same underlying principles are at work and that *S. mediterranea* is therefore a suitable model system for probing the physiological basis of Kleiber's law.

## Size-dependence of planarian growth/degrowth dynamics

The physiological causes of planarian body size fluctuations are growth and degrowth. Therefore, understanding their underlying regulation might provide insights into the size-dependence of the metabolic rate. Planarian body size measurements are challenging due to their soft and highly deformable bodies. We therefore adapted our semi-automated live-imaging pipeline that extracts size measurements from multiple movie frames displaying the same animal in an extended body posture (*Werner et al., 2014*). We found that plan area provides the most robust, non-lethal size measure (*Figure 2—figure supplement 1* and (*Werner et al., 2014*)), which we therefore use in the following. One first important question was to what extent organismal size changes reflect a change in cell number. Since previous cell number estimates produced conflicting results (*Romero and Baguñà, 1991*; *Takeda et al., 2009*) we developed two independent assays. First, we combined single-animal dissociation into individual cells (*Romero and Baguñà, 1991*) with automated counting of fluorescently stained nuclei (*Figure 2A*, *top* and *Figure 2—figure supplement 2*). Second, we used quantitative Western blotting to quantify the amount of the core Histone H3 in lysates of individual worms, which we found to increase linearly with the number of FACS-sorted cells (*Figure 2A*, *bottom*). Applying both assays to individually sized *S. mediterranea* revealed a close agreement between the two methods and scaling of cell numbers with plan area by a power law with the exponent 1.19 (*Figure 2B*). These data are consistent with previous conclusions that planarian body size changes predominantly reflect changes in cell number rather than cell size (*Baguñà et al., 1990*).

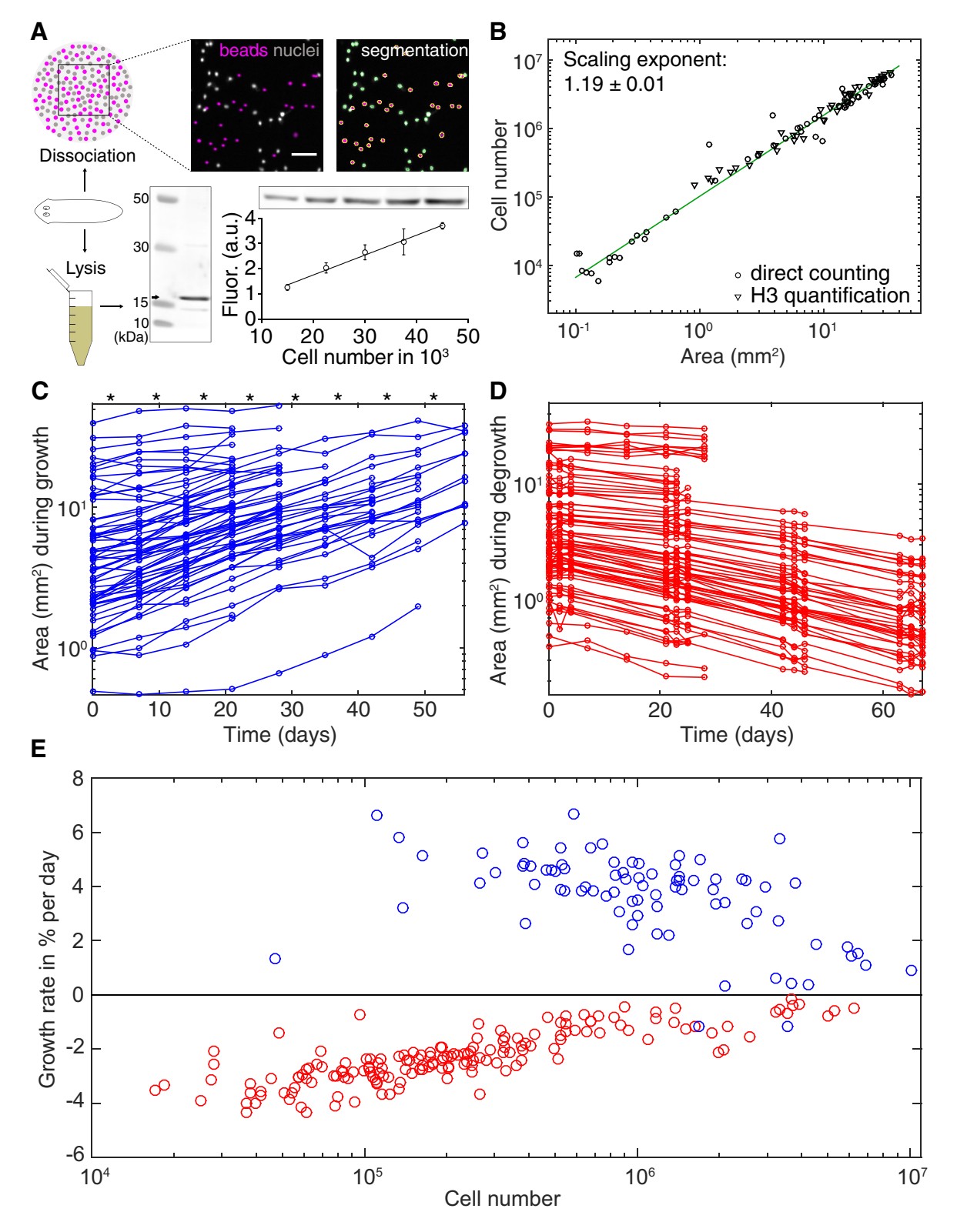

**Figure 2.** Growth and degrowth dynamics in *S.mediterranea*. (A) Assays to measure organismal cell numbers. (Top) image-based quantification of nuclei (grey) versus tracer beads (magenta) following whole animal dissociation in presence of the volume tracer beads. (Bottom) Histone H3 protein quantification by quantitative Western blotting, which scales linearly with the number of FACS-sorted cells (bottom right). The line represents a fitted linear regression (data of 4 technical replicates) and serves as standard for converting the H3 band in planarian lysates (bottom left) run on the same gel

*Figure 2 continued on next page*

*Figure 2 continued*

into cell numbers. Values are shown as mean ± standard deviation. (B) Organismal cell number versus plan area scaling, by nuclei counts (circles) or Histone H3 protein amounts (triangles) (see also *Figure 2—figure supplements 1* and *2*). The scaling exponent ± standard error was derived from a linear fit and represents the exponent *b* of the power law $y = ax^b$. Each data point represents one individual animal and the mean of several technical replicates, Histone H3 method: nine independent experiments including five animals each; image-based approach: four independent experiments including 18, 10, 10 and 12 animals each. See *Figure 2—source datas 1–3* for numerical data. (C) Plan area changes of individual animals during growth. * indicate feeding time points (1x per week). (D) Plan area change of individual animals during degrowth. (E) Size-dependence of growth (blue) and degrowth rates (red) (see also *Figure 2—figure supplement 3A*). Individual data points were calculated by exponential fits to traces in (C) and (D) (growth: two overlapping time windows, degrowth: three overlapping time windows) and using the cell number/area scaling law from (B) to express rates as % change in cell number/day. The positive growth rates and negative degrowth rates are plotted on the same axis to facilitate comparison of size dependence. See *Figure 2—source data 5* for data of (C) and (D).

DOI: https://doi.org/10.7554/eLife.38187.006

The following source data and figure supplements are available for figure 2:

**Source data 1.** Numerical data cell number measurements.
DOI: https://doi.org/10.7554/eLife.38187.012
**Source data 2.** Raw numerical data Histone H3 method (quantitative Western blotting).
DOI: https://doi.org/10.7554/eLife.38187.013
**Source data 3.** CellProfiler results tables image-based approach.
DOI: https://doi.org/10.7554/eLife.38187.014
**Source data 4.** MATLAB code for extraction of planarian body size.
DOI: https://doi.org/10.7554/eLife.38187.015
**Source data 5.** Numerical data growth/degrowth.
DOI: https://doi.org/10.7554/eLife.38187.016
**Figure supplement 1.** Measurement of planarian body size.
DOI: https://doi.org/10.7554/eLife.38187.007
**Figure supplement 1—source data 1.** Numerical data for *Figure 2—figure supplement 1D and E*.
DOI: https://doi.org/10.7554/eLife.38187.008
**Figure supplement 2.** Validation of image-based quantification of organismal cell number.
DOI: https://doi.org/10.7554/eLife.38187.009
**Figure supplement 2—source data 1.** CellProfiler pipeline, numerical data, raw images and segmentation for validation of image-based cell counting.
DOI: https://doi.org/10.7554/eLife.38187.010
**Figure supplement 3.** Degrowth rates are independent of feeding history.
DOI: https://doi.org/10.7554/eLife.38187.011

Further, knowledge of the cell number/area scaling law allows the accurate interconversion of plan area into cell numbers in the experiments below.

To measure growth and degrowth rates, we quantified the change in plan area of individual *S. mediterranea* subjected to feeding at regular time intervals (*Figure 2C*) or continuous starvation (*Figure 2D*). Although individual measurements were noisy due to the aforementioned size quantification challenges, the data on > 100 animals cumulatively reveal that the growth rate of *S. mediterranea* decreases with body size, consistent with previous data (*Figure 2E* and *Figure 2—figure supplement 3A*; *Baguñà et al., 1990*). Unexpectedly, our analysis additionally revealed a similar size dependence of the degrowth rate. Interestingly, the degrowth rates appeared to be generally independent of feeding history and thus primarily a function of size (*Figure 2—figure supplement 3B*). Taken together, our findings demonstrate that not only the specific metabolic rate (*Figure 1C–D*), but also the growth/degrowth rates decrease with body size in *S. mediterranea*.

## Systems-level control of planarian growth/degrowth dynamics

Since growth reflects the metabolic assimilation of environmental resources and degrowth their subsequent catabolism, both are related to the overall metabolic rate of the animal. Consequently, the size dependence of growth/degrowth (*Figure 2E*) and metabolic rate (*Figure 1C–D*) might reflect a common physiological origin of the underlying scaling laws in planarian energy metabolism. We therefore devised a theoretical framework of planarian growth/degrowth as a function of the metabolic energy budget (*Figure 3A*). The central element of our model and previous approaches (*Hou et al., 2008*; *Kooijman, 2009*) is the organismal energy content *E*, which represents the sum of all physiologically accessible energy stores (e.g., carbohydrates, lipids and proteins). The energy

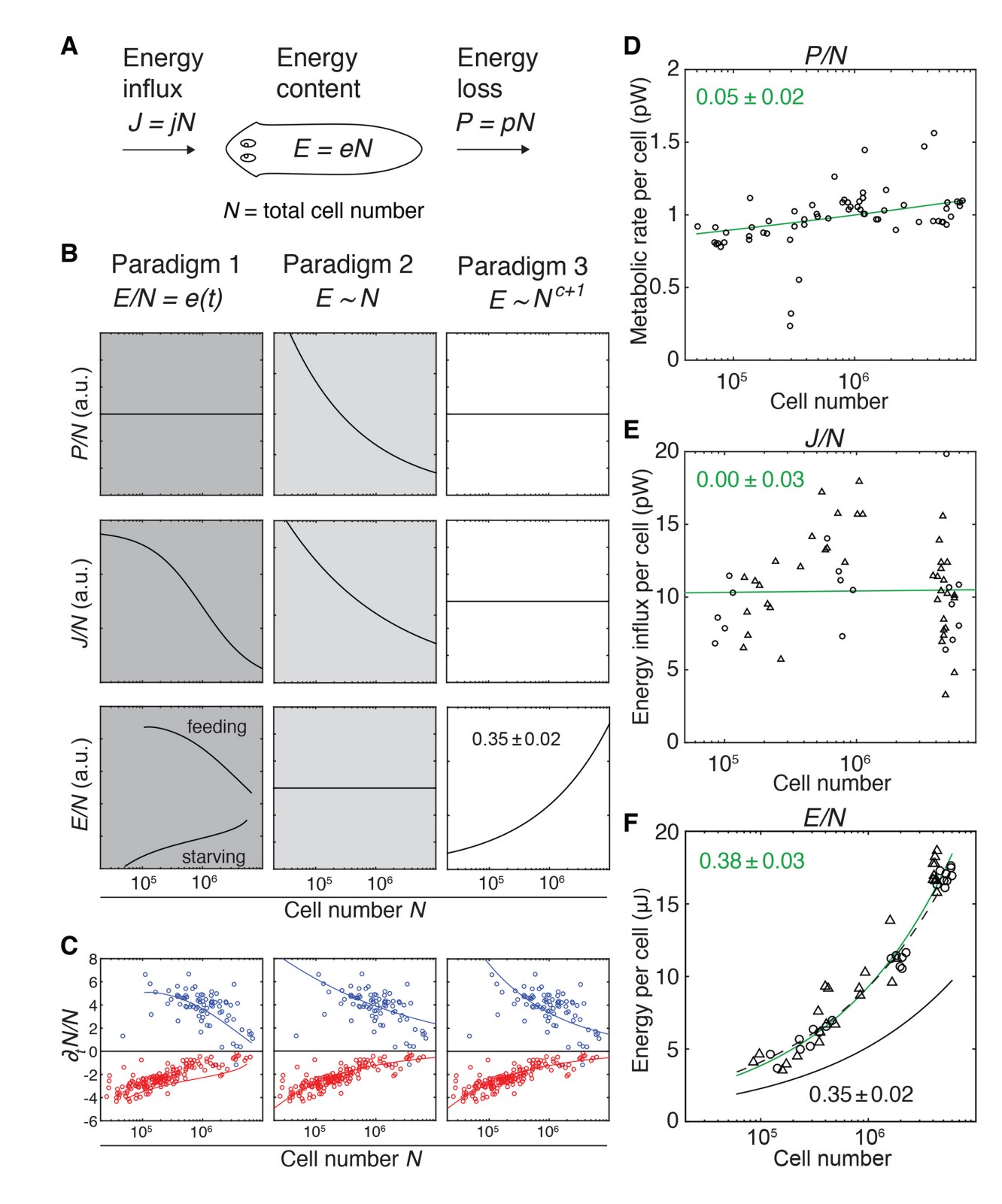

**Figure 3.** Size-dependent scaling of energy content explains growth/degrowth dynamics. (A) Planarian energy balance model. At the organismal level, changes in the physiological energy content $E$ result from a change in the net energy influx $J$ (feeding) and/or heat loss $P$ (metabolic rate). Dividing $E$, $J$ and $P$ by the total cell number $N$ approximates the energy balance on a per-cell basis. (B) Three hypothetical control paradigms of $E$ during growth and degrowth (columns), which make specific predictions regarding the size-dependence of $J/N$, $E/N$ and $P/N$ (rows). Prediction traces and scale exponents

*Figure 3 continued on next page*

*Figure 3 continued*

were generated by modelling the measured growth/degrowth rates (*Figure 2E*) with the indicated control paradigm assumptions (see also Appendix 1 and *Figure 3—figure supplement 1*). (C) Fit of the three control paradigms to the measured growth/degrowth rates (*Figure 2E*). (D) Metabolic rate per cell ($P/N$) versus organismal cell number ($N$). Data points were derived by conversion of the measurements from the metabolic rate/dry mass scaling law (*Figure 1—figure supplement 1B*) via the measured cell number/plan area (*Figure 2B*) and plan area/dry mass conversion laws (*Figure 3—figure supplement 2A*). The scaling exponent ± standard error was derived from the respective linear fit (green line) and represents the exponent $b$ of the power law $y = ax^b$. (E) Energy influx per cell versus organismal cell number ($N$). Data points reflect single-animal quantifications of ingested liver volume per plan area as shown in *Figure 3—figure supplement 2B–D*, converted into energy influx/cell using the plan area/cell number scaling law (*Figure 2B*) and the assumption that 1 µl of liver paste corresponds to 6.15 J (*USDA Agricultural Research Service, 2016*; *Overmoyer et al., 1987*). Circles, 2 weeks starved and triangles, 3 weeks starved animals. The scaling exponent ± standard error was derived from linear fits (green line) and represents the exponent $b$ of the power law $y = ax^b$. (F) Energy content per cell ($E/N$) versus organismal cell number ($N$). Data points reflect bomb calorimetry quantifications of heat release upon complete combustion of size matched cohorts of known dry mass as shown in *Figure 3—figure supplement 2E*, converted via the measured cell number/plan area (*Figure 2B*) and plan area/dry mass conversion laws (*Figure 3—figure supplement 2A*). Circles, 1 week starved and triangles, 3 weeks starved animals. The scaling exponent ± standard error was derived from a linear fit (green line) to the data and represents the exponent $b$ of the power law $y = ax^b$. Solid black line, prediction from model three for the physiological energy content per cell assuming a constant metabolic rate $P/N = 1$ pW. Dashed line corresponds to respective prediction under the assumption that the physiological energy (solid black line) amounts to 50% of combustible gross energy in the animal. See *Figure 3—source data 1* for numerical data of (C)-(F).

DOI: https://doi.org/10.7554/eLife.38187.017

The following source data and figure supplements are available for figure 3:

**Source data 1.** Numerical data for *Figure 3*.
DOI: https://doi.org/10.7554/eLife.38187.021
**Figure supplement 1.** Further explanation of model paradigm 1.
DOI: https://doi.org/10.7554/eLife.38187.018
**Figure supplement 2.** Validation of model paradigms.
DOI: https://doi.org/10.7554/eLife.38187.019
**Figure supplement 2—source data 1.** Numerical data for *Figure 3—figure supplement 2*.
DOI: https://doi.org/10.7554/eLife.38187.020

content $E$ fuels all metabolic processes within the animal, which collectively convert $E$ into heat that we can experimentally measure by our microcalorimetry approach (*Figure 1C–D*). Hence, starvation reduces the energy content $E$ via net catabolism and degrowth. However, $E$ increases if the influx of energy obtained from the food $J$ exceeds the energy lost through heat $P$, which leads to net assimilation of resources and thus growth. The fact that planarians grow/degrow largely by a change in total cell numbers (*Figure 2B*) (*Baguñã et al., 1990*; *Romero and Baguñà, 1991*), further fundamentally interconnects the organismal energy balance with organismal cell numbers. While excess food energy intake stimulates increased cell proliferation (*Baguñà, 1974*) and growth, the starvation-induced net loss of energy manifests in a decrease of total cell numbers and thus, body size. Therefore, our framework relates changes in cell number during growth/degrowth to the energy content of the animal (*Figure 3A*). Importantly, our model does not make any assumptions regarding the underlying cellular or metabolic mechanisms, but simply states the physical energy balance of planarians.

With our quantitative growth/degrowth data as experimental constraint (*Figure 2C–E*), the model allows us to explore hypothetical systems-level control paradigms of growth/degrowth dynamics and thus different potential origins of the observed size-dependencies in our data (see also Appendix 1). The first paradigm (*Figure 3B*, left column) assumes dynamic changes in the organismal energy content depending on feeding conditions and changes in cell number (e.g., rates of cell division and/or cell death) depending on the energy content per cell (*Figure 3—figure supplement 1*). Consequently, two planarians with the same cell number might have different energy levels depending on the respective feeding history. In paradigm 2 (*Figure 3B*, centre column), the energy content remains always proportional to total cell number, that is it scales isometrically. Thus, growth occurs when 'surplus' energy obtained from food intake is converted into new cells, whereas degrowth is the consequence of catabolism of existing cells in order to replenish metabolic energy. In paradigm 3, the energy content is also tightly coupled to cell number, but scales in a size-dependent manner with a characteristic exponent $c + 1$, that is it scales allometrically with total cell number (*Figure 3B*, right column). Although more complex scenarios are possible, the three paradigms cover the three principal possibilities of $e = E/N$ as dynamic (paradigm 1), size-invariant (paradigm 2) or size-

dependent variable (paradigm 3). Theoretical analysis reveals that the measured growth/degrowth dynamics can be fit with all three paradigms (*Figure 3C*), thus demonstrating their principal feasibility as systems-level control principles. However, the paradigms differ in their specific predictions of the scaling behaviours of the metabolic rate $P$ and energy influx $J$ with organismal cell number $N$ (*Figure 3B*).

To experimentally distinguish between the paradigms, we therefore quantified the energy loss via the metabolic rate $P$, food energy influx $J$ and the energy content $E$ as a function of organismal cell number ($N$). In order to obtain values for $P/N$ (metabolic rate/cell), we converted our measurements of $P$ as a function of dry mass (*Figure 1—figure supplement 1B*) using the scaling laws for $N$ and dry mass with plan area (*Figure 2B* and *Figure 3—figure supplement 2A*). As shown in *Figure 3D*, the $P/N$ estimates are of the order of 1 pW, similar to the average metabolic rate of a human cell (*Bianconi et al., 2013*; *Purves and Sadava, 2004*). Further, $P/N$ is essentially independent of organismal cell number and animal size (scale exponent 0.05 ± 0.02), which rules out paradigm 2 (*Figure 3B*) as possible control principle. The size *independence* of $P/N$ is further intriguing, as it implies that the size *dependence* of $P/M$ as foundational basis of Kleiber's law originates from size dependencies of $M/N$ (mass per cell; see below).

To estimate the energy influx $J$, we developed an assay based on the homogenous dispersion of a known amount of small fluorescent beads in a known volume of planarian sustenance food (liver paste). Lysis of pre-sized animals immediately after feeding and quantification of bead numbers in the lysate thus provided a measure of the ingested food volume as a function of size (*Figure 3—figure supplement 2B–D*). Although individual measurements varied significantly (likely reflecting inter-animal differences under our *ad libitum* feeding conditions), the energy influx per cell, $J/N$, did not display a clear size dependence (exponent of 0.00 ± 0.03) (*Figure 3E*). Therefore, the volume of ingested food and thus energy influx remains proportional to organismal cell number across the entire size range, which argues against both paradigms 1 and 2 (*Figure 3B*).

To approximate the energy content $E$ of entire worms, we turned to bomb calorimetry. This method quantifies the heat release upon complete combustion of dried tissue in pure oxygen, thus providing a measure of gross energy content (*McDonald, 2002*). Our assay conditions allowed reproducible quantification of $E$ of as little as 3 mg of dried tissue (*Figure 3—figure supplement 2E*), corresponding to a cohort of 200 planarians with a length of 2 mm (*Figure 3—figure supplement 2A* and *Figure 2—figure supplement 1E*). Intriguingly, the energy content per cell, $E/N$, significantly increased with organismal cell numbers (scaling exponent 0.38 ± 0.03, *Figure 3F*), as assumed by paradigm 3 (*Figure 3B*). Moreover, the experimentally measured scaling exponent agrees quantitatively with the prediction of paradigm 3 (physiologically accessible energy) on basis of the experimentally measured growth/degrowth rates (*Figure 3F*; black solid line). The experimentally measured gross energy content and the physiologically accessible energy content $E$ obtained from model 3 (green and black solid lines in *Figure 3F*) differ by a constant factor of about two, irrespective of feeding history. This is consistent with the previously inferred size- rather than feeding history dependence of the organismal degrowth rate (*Figure 2—figure supplement 3B*). The fact that the scaling exponent follows the prediction of paradigm three demonstrates the quantitative agreement between model and experiment and identifies size-dependent energy storage as systems-level control paradigm of planarian growth/degrowth dynamics.

## Size dependence of physiological energy stores

Since biological systems store energy in the form of biochemical compounds, size-dependent energy storage should consequently result in changes in the biochemical composition of planarians. Little is currently known about planarian energy metabolism, but animals generally store metabolic energy in the form of triglycerides (TGs) inside lipid droplets (*Birsoy et al., 2013*) or glucose in the form of glycogen granules (*Roach et al., 2012*). We first stained cross-sections of large and small animals with the lipid droplet marker LD540 (*Spandl et al., 2009*). Both revealed prominent lipid droplets primarily within the intestinal epithelium, thus suggesting that the planarian intestine serves as a fat storage organ, similar to the *C. elegans* intestine (*Mak, 2012*). However, the amount and size of the droplets per intestinal cell notably increased in large animals (*Figure 4A*). To obtain a quantitative measure of the size-dependence of the lipid content, we optimized total lipid extraction for planarians (*Figure 4—figure supplement 1A*) and used mass spectrometry to measure the absolute amounts of various lipid classes (*Figure 4—figure supplement 1B*). The 88-fold increase in TGs per unit cell in

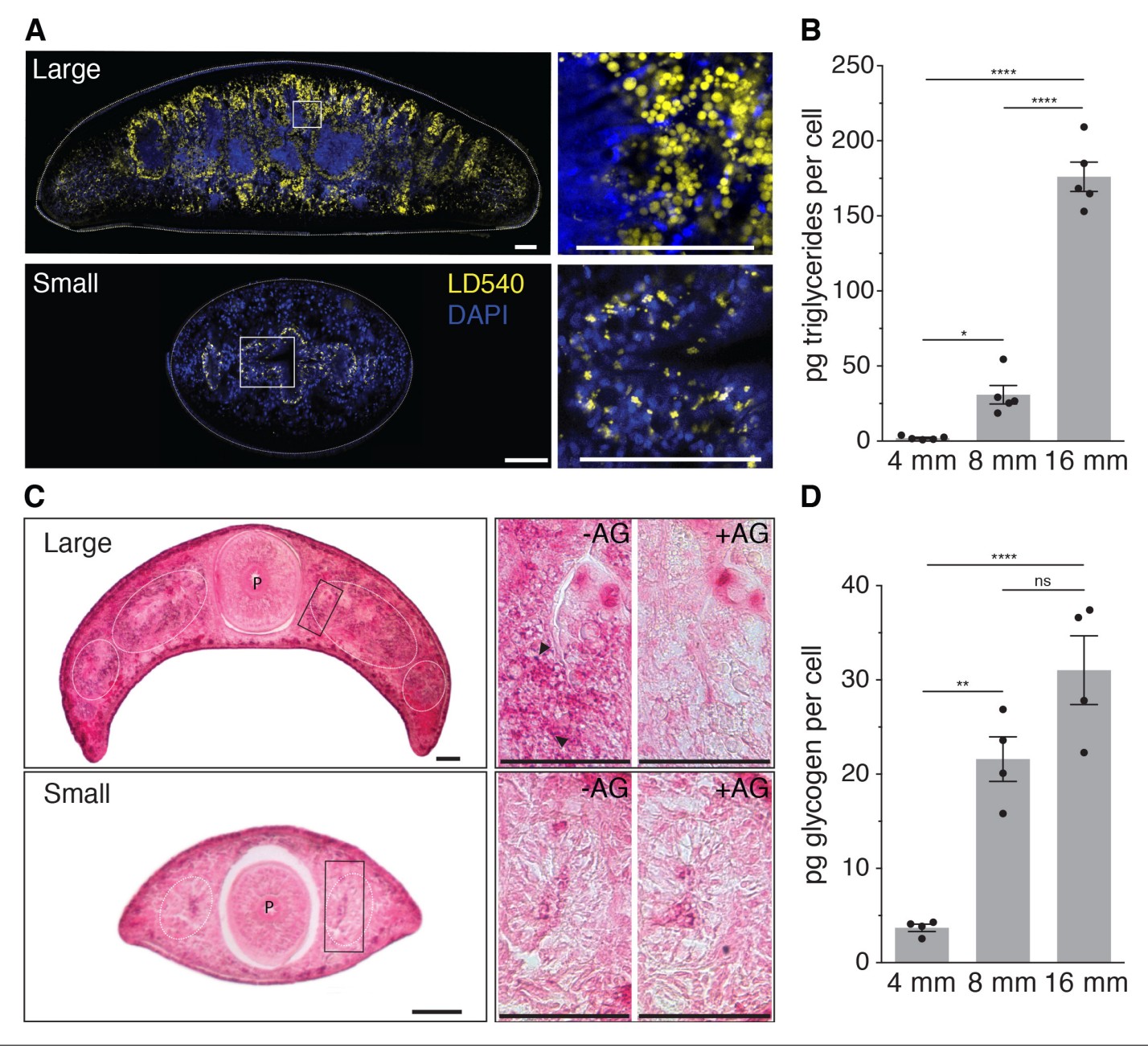

**Figure 4.** Size-dependence of lipid and glycogen storage. (**A**) Lipid droplet (LD540, yellow) (*Spandl et al., 2009*) and nuclei (DAPI, blue) staining of pre-pharyngeal transverse cross sections of a large (16 mm length, top left) and a small (4 mm, bottom left) planarian. Right, magnified view of the boxed areas to the left. Scale bars, 100 μm. See *Figure 4—source data 1* for raw images. (**B**) Mass spectrometry-based quantification of triglycerides in animals of the indicated size (*Figure 4—figure supplement 1A–B*). All values were normalized to organismal cell numbers using the previously established length versus area (*Figure 2—figure supplement 1E*) and *N/A* (*Figure 2B*) scaling laws. Bars mark mean ± SEM. n = 5 biological replicates consisting of 40 pooled 4 mm, 20 8 mm and 6 16 mm long animals analysed in two technical replicates. Significance assessed by one-way ANOVA, followed by Tukey's post-hoc test (*$p_{adj}$ ≤ 0.05, ****$p_{adj}$ ≤ 0.0001). See *Figure 4—source data 2* for numerical data and statistics. (**C**) Histological glycogen staining (Best's Carmine method) of pharyngeal transverse cross sections of a large (16 mm, top left) and a small (4 mm, bottom left) planarian. White circles: outline of intestine branches. P: Pharynx. Right, magnified view of the boxed areas to the left (black rectangles).+AG, pre-treatment with amyloglucosidase, which degrades glycogen; -AG, no pre-treatment of adjacent section. Arrow heads point to small, densely staining glycogen granules. Scale bars, 100 μm. See *Figure 4—source data 1* for raw images. (**D**) Quantification of organismal glycogen content using an enzyme-based colorimetric assay in animals of the indicated length (*Figure 4—figure supplement 1D–F*). Bars mark mean ± SEM. n = 4 biological replicates (independent experiments), 40 pooled 4 mm, 20 8 mm, 8 16 mm analysed in three technical replicates. Significance assessed by one-way

*Figure 4 continued on next page*

*Figure 4 continued*

ANOVA, followed by Tukey's post-hoc test (ns not significant, **$p_{adj} \leq 0.01$, ****$p_{adj} \leq 0.0001$). See *Figure 4—source data 2* for numerical data and statistics.

DOI: https://doi.org/10.7554/eLife.38187.022

The following source data and figure supplement are available for figure 4:

**Source data 1.** Raw images lipid droplet and glycogen.

DOI: https://doi.org/10.7554/eLife.38187.024

**Source data 2.** Raw data lipid mass spectrometry, glycogen assay and statistics tables.

DOI: https://doi.org/10.7554/eLife.38187.025

**Figure supplement 1.** Assays for lipid and glycogen quantification in planarians.

DOI: https://doi.org/10.7554/eLife.38187.023

large planarians as compared to small animals (*Figure 4B*) demonstrates a striking size dependence of lipid stores in *S. mediterranea*.

To assess a possible size dependence of carbohydrate stores, we applied Best's Carmine stain to cross-sections of large and small animals in order to visualize glycogen granules (*Figure 4C*, left). With adjacent sections pre-treated with the glycogen degrading enzyme amyloglucosidase as specificity control (*Figure 4C*, right), we detected specific staining in the intestine. Together with the likewise intestine-enriched expression of glycogen synthesis genes (*Figure 4—figure supplement 1C*), this result emphasizes the organ's likely central role in energy homeostasis. Interestingly, also the intensity of glycogen staining appeared stronger in large animals (*Figure 4C*, right) and the quantification of glycogen content in animal extracts by an enzyme-based assay (*Figure 4—figure supplement 1D–F*) demonstrated a > 8 fold increase in the amount of glycogen/cell in large over small animals (*Figure 4D*). Therefore, both the lipid and carbohydrate stores of *S. mediterranea* are strongly size-dependent, which conclusively confirms our model's prediction of size-dependent energy storage as a systems-level control paradigm of planarian growth and degrowth.

## Energy reserves and cell number govern Kleiber's law in planarians

The size-dependent increase in the mass of lipid and glycogen stores is intriguing also in light of the previous indications that Kleiber's law in planarians might originate from a size-dependent increase in mass per cell, rather than a decrease in metabolic rate (*Figure 3D*). To explore this potential link between the regulation of growth dynamics and Kleiber's law, we first investigated the relative contributions of mass-cell number allometries to the emergence of the 3/4 exponent. As a direct test, we derived the size dependence of cell numbers versus mass, using the various scaling laws established during the course of this study. As shown in *Figure 5A*, cell numbers scale with wet and dry mass with scale exponents of $0.72 \pm 0.01$ and $0.74 \pm 0.01$, respectively. This demonstrates that the mass per cell indeed increases disproportionately with size and with a very similar scaling exponent as for Kleiber's law (*Figure 1C*). In conjunction with the practically proportional scaling of cell number and metabolic rate (*Figure 5B*, scaling exponent $0.96 \pm 0.02$), these data demonstrate conclusively that the 3/4 exponent of the metabolic rate/mass scaling law derives from the underlying scaling law of mass/cell.

To quantitatively assess the contributions of energy stores to the mass/cell scaling exponent and thus to Kleiber's law, we analysed the composition of the dry mass in small, medium and large animals. In addition to storage lipids and glycogen, we quantified total protein (*Figure 5—figure supplement 1A*), non-glycogen carbohydrates (*Figure 5—figure supplement 1B–C*) and other polar and non-polar lipids (*Figure 4—figure supplement 1B*). In comparison with the 8- and 88-fold increase of glycogen and triglyceride contributions to the dry mass/cell increase, the relative contribution of protein, other polar/non-polar lipids and non-glycogen carbohydrates were less variable between small and large animals (*Figure 5C*). Our quantitative assays further allowed us to assess the absolute mass contribution of each compound class to the size-dependent dry mass increase and thus to the origins of the 3/4 exponent. Intriguingly, the latter was largely explained by the mass of triglycerides and glycogen, with additional minor contributions from other carbohydrates, polar/non-polar lipids and protein (*Figure 5C*). Overall, our results therefore demonstrate that size-dependent energy storage causes Kleiber's law scaling in *S. mediterranea*.

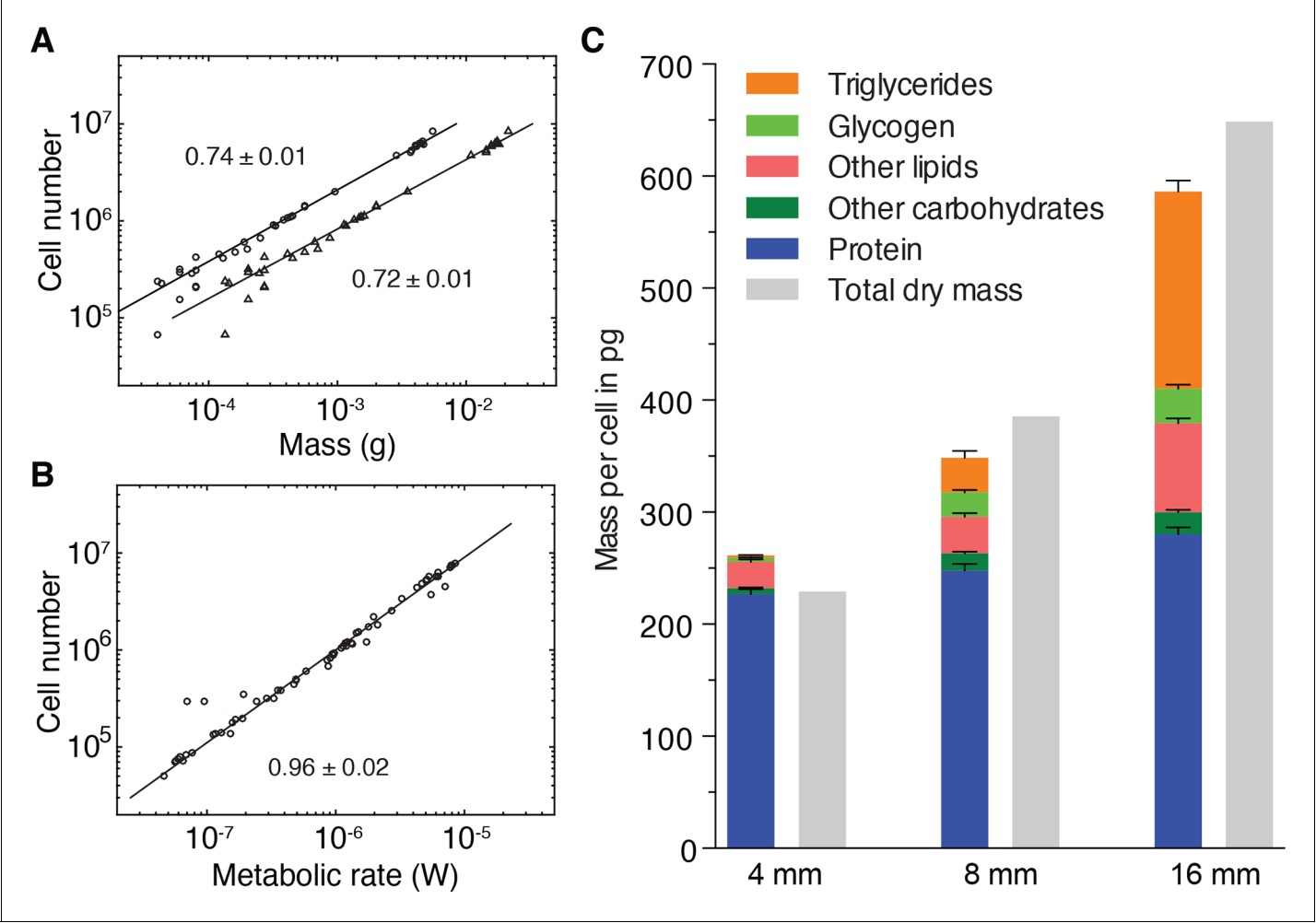

**Figure 5.** Size-dependent energy storage explains Kleiber's law scaling. (A) Cell number versus dry mass (circles) or wet mass (triangles) based on the data from **Figure 3—figure supplement 2A**. Cell numbers were converted from area using the *N/A* scaling law (**Figure 2B**). Dry and wet mass conversion is given by **Figure 1B**. Scaling exponents ± standard errors were derived from respective linear fits and represent the exponent *b* of the power law $y = ax^b$ (B) Cell number versus metabolic rate, derived from **Figure 1C** with scaling laws of **Figure 2B** and **Figure 3—figure supplement 2A**. The scaling exponent ± standard error was derived from respective linear fits and represents the exponent *b* of the power law $y = ax^b$. (C) Mass composition (coloured) and total dry mass (grey) per cell in animals of the indicated body length. Triglyceride and glycogen measurements are taken from **Figure 4B and D**, respectively. Quantification of other (polar and non-polar) lipids is based on the mass-spectrometry data from **Figure 4B** (see also **Figure 4—figure supplement 1B**) (n = 5 biological replicates; $p_{adj}$ = 0.1720 (no significance) 8 vs. 4 mm, $p_{adj}$ < 0.0001 16 vs. 4 mm, $p_{adj}$ < 0.0001 16 vs. 8 mm; two technical replicates). Other carbohydrates represent total carbohydrate minus glycogen. n = 4 biological replicates (independent experiments), 40 pooled 4 mm, 20 8 mm, 8 16 mm long animals; $p_{adj}$ = 0.0047 8 vs. 4 mm, $p_{adj}$ = 0.0005 16 vs. 4 mm, $p_{adj}$ = 0.2790 16 vs. 8 mm; three technical replicates. Protein content was measured colorimetrically. n = 4 biological replicates (independent experiments), 44 pooled 4 mm, 10 8 mm, 10 16 mm long animals; $p_{adj}$ = 0.0020 8 vs. 4 mm, $p_{adj}$ < 0.0001 16 vs. 4 mm, $p_{adj}$ = 0.0007 16 vs. 8 mm) (see also **Figure 5—figure supplement 1**). Significance was assessed by one-way ANOVA followed by Tukey's post-hoc test. All values were normalised to the total cell number using the previously established length-area (**Figure 2—figure supplement 1E**) and *N/A* (**Figure 2B**) scaling laws. Total dry mass was independently measured (**Figure 3—figure supplement 2A**) and correlated with length using the length-area relationship (**Figure 2—figure supplement 1E**). All values are shown as mean ± SEM. See **Figure 5—source data 1** for numerical data and statistics.

DOI: https://doi.org/10.7554/eLife.38187.026

The following source data and figure supplement are available for figure 5:

**Source data 1.** Raw data and statistics tables for measurement of other lipids, carbohydrates and protein.
DOI: https://doi.org/10.7554/eLife.38187.028

**Figure supplement 1.** Validation of protein and total carbohydrate quantifications.
DOI: https://doi.org/10.7554/eLife.38187.027

## Discussion

Given the practically universal prevalence of Kleiber's law amongst animals (*Figure 1D*), our finding that asexual *S. mediterranea* also display 3/4 scaling of metabolic rate with mass may seem hardly surprising. However, the physiological processes that bring about mass changes in planarians are very unusual. Planarians grow when fed and the magnitude of the growth is tremendous as shown here, amounting to a > 40 fold increase in body length (*Figure 2—figure supplement 1E*) and a corresponding > 800 fold increase in organismal cell numbers (*Figure 2B*) or > 9 000-fold increase in dry weight (*Figure 2—figure supplement 1E* and *Figure 3—figure supplement 2A*). The *S. mediterranea* growth range thus by far exceeds the < 5 fold length and ~70 fold weight post-birth growth of blue whales (*Sears and Perrin, 2009*) and quantitatively approaches the tremendous post larval length and weight growth of some fish species (*Field-Dodgson, 1988*; *Mills et al., 2004*). However, what is rather unique about planarian growth is that it is entirely reversible. Starving animals literally shrink by a progressive reduction of total cell numbers and the likely catabolism of the surplus cells (*Baguñà et al., 1990*; *Rink, 2018*). As a result, the adult body size of planarians fluctuates continuously within the above size interval as a function of food supply. Although our data do not explicitly address the contribution of cell size changes, they confirm changes in cell numbers as the predominant mechanism of planarian growth/degrowth dynamics (*Baguñà et al., 1990*). Planarians continuously turnover all organismal cell types via the division progeny of their abundant adult pluripotent stem cells and the death of differentiated cells (*Baguñà, 2012*; *Rink, 2018*) and growth/degrowth therefore necessarily involve dynamic adjustments of the respective rate constants (see below). Planarian growth processes are therefore highly unusual as compared to other animals, both in terms of the bidirectionality of growth and the pivotal involvement of adult pluripotent stem cells. This makes the applicability of Kleiber's law to planarian growth/degrowth actually remarkable and emphasizes its underlying reflection of a fundamental and still largely mysterious size dependence of animal metabolism.

A priori, the 3/4 power scaling of metabolic rate with mass signifies a systematic change in the ratio between the two parameters, which is commonly assumed to be driven by a specific decrease of the cellular metabolic rate with increasing mass (*Smith, 1956*, *Kunkel et al., 1956*, *Jansky, 1961*; *Jansky, 1963*, *Maino et al., 2014*; *West et al., 1997*), see below). Our finding that the average metabolic rate/cell in starving *S. mediterranea* is size-independent (*Figure 3D*) was consequently somewhat surprising. Although our microcalorimetry measurements cannot rule out size-dependent changes in metabolic networks, the constant ratio between the experimentally measured gross energy content and the modelled net energy content over the entire size range (*Figure 3F* and *Figure 3—figure supplement 2E,F*) entails a constant food assimilation efficiency and thus likely also a size-independence of the underlying metabolic networks. Further, our results conclusively identify a specific increase in the mass/cell (*Figure 5A*) due to size-dependent energy storage (*Figure 5C*) as the sole cause of the 3/4 power scaling relationship in starving *S. mediterranea*. With the additional interplay of growth/degrowth dynamics, our results implicate a metabolic trade-off in the physiological origins of Kleiber's law in planarians: While small planarians grow rapidly due to a predominant 'investment' of ingested food into metabolically active new cells rather than inert energy stores, large planarians grow more slowly due to a predominant investment in metabolically inert energy stores rather than new cell production.

One of the important questions raised by our results is whether size-dependent energy storage is cause or consequence of size dependent growth dynamics. It is conceivable that energy storage is causal, for example via a size-dependent increase in the rate of lipid sequestration by intestinal cells and a consequent decrease in lipid availability as hypothetical fuel source for stem cell proliferation. Alternatively, it is possible that a size-dependent mechanism limits the fraction of proliferating stem cells and that intestinal lipid accumulation represents a secondary consequence of decreased lipid catabolism elsewhere in the animal. These considerations further highlight the need for a quantitative analysis of the planarian feeding response. Planarians grow because of the rapid and transient upregulation of stem cell divisions in response to food intake, which translates into a burst of progenitor production, an increase of total organismal cell numbers and thus a growth pulse at the organismal level (*Baguñà, 1974*). Since all metabolic rate quantifications in this study were limited to 2–3 weeks starved animals, they cannot inform on the actual growth phase. It is therefore conceivable that the metabolic rate/cell soon after feeding might in fact display size-dependencies, for

example due to a decrease in the fraction of stem cells entering S-phase. Such a scenario might additionally reconcile the size-dependence of the growth rate (*Figure 2E*) with the size-independence of the average cellular metabolic rate in starving animals (*Figure 3D*) and quantifications of the size-dependence of stem cell dynamics and the metabolic rate during the feeding response will be further important aspects of understanding the mechanistic origins of the 3/4 exponent in planarians.

Size-dependent energy storage as the physiological basis of Kleiber's law in planarians raises the question whether the same principle might also apply in other animals. The deep evolutionary conservation of lipid storage and other aspects of core energy metabolism (*Birsoy et al., 2013*), the much-reduced specific metabolic rate of adipose as compared with other tissues (*Elia, 1992*; *Wang et al., 2010*) and the allometric scaling of fat content with body mass across a wide range of vertebrates (*Calder, 1984*; *Pitts and Bullard, 1968*; *Prothero, 1995*) are principally compatible with a broad applicability of our findings. However, as exemplified by human dieting, the mass fraction of vertebrate energy stores tends to fluctuate tremendously over time due to feeding history dependent energy storage (e.g., paradigm one in our models, *Figure 3B*). Additionally, mammalian fat storage has a strong genetic component and per unit weight, arctic species tend to have a higher fat content than similar-sized species from temperate climate zones (*Blix, 2016*). Although systematic mass dependencies of lipid stores may therefore be difficult to detect in indiscriminate inter-species comparisons, a quantitative investigation of intra-species metabolic rate/mass scaling phenomena might also provide interesting insights (*Glazier, 2005*). Further, it is important to stress that even in planarians, lipids are not the sole cause of the mass/cell allometry. Glycogen and proteins also contribute (*Figure 5C*) and it is thus plausible that other metabolically inert compounds or combinations of compounds might drive the 3/4 mass/cell allometry in other species. In fact, assuming the general size independence of metabolic rate/cell and a 3/4 power law dependence of mass/cell predicts that an adult human of 70 kg should consist of $6-20 \times 10^{13}$ cells (*Purves and Sadava, 2004*) (see Appendix 2). Interestingly, the current experimental estimate of $3.7 \times 10^{13}$ cells/70 kg adult human (*Bianconi et al., 2013*) comes close to this value, thus indicating that the 3/4 power law scaling of mass/cell that accounts for Kleiber's law in planarians could also apply to other taxonomic groups. This in turn might ultimately root Kleiber's law in the size-dependence of a food assimilation trade-off between metabolically active versus inert biomass and the elucidation of the size dependent lipid storage mechanisms in planarians may prove informative in this respect. Moreover, the hypothesis that mass allometries rather than metabolic rate allometries generally account for Kleiber's law might be interesting to explore in other species.

Finally, this leaves the mystery of why the scale exponent of metabolic rate with mass in animals is always 3/4. The main approach to this problem so far has been physical theories. Interestingly, our demonstration of a trade-off between energy storage and growth rate in planarians converges on a central premise of the Dynamic Energy Budget (DEB) theory, which is one of the well-known theoretical explanations of Kleiber's law. The DEB theory derives the 3/4 exponent out of the assumption of surface-limited energy store mobilization (*Kooijman, 2009*; *Maino et al., 2014*). Briefly, the DEB theory divides organismal mass into interconvertible reserve (metabolically inert energy stores) and structural mass (metabolically active cell mass) and assumes isometric scaling of the two compartments with body size (e.g., constant ratio of compartment diameters). The decreasing surface-to-volume ratio with increasing size limits energy retrieval from the storage compartment and thus ultimately metabolic rate. Although the mobilization of triglycerides from lipid droplets is indeed surface-limited (*Walther and Farese, 2012*), observations that conflict with the DEB theory include the size-independence of the metabolic rate during starvation in planarians and generally the variable number and size of lipid droplets (e.g., *Figure 4A*). A second prominent physical theory is the so-called WEB theory that envisages the origins of the 3/4 exponent in the intrinsic transport capacity limitations of space-filling fractal networks (*West et al., 1997*). Although planarians lack the vascular or alveolar networks that are commonly assumed to constitute the anatomical basis of the WEB theory, they assimilate and distribute metabolic energy via the branched tubular network of their intestine (termed 'gastrovasculature', *Forsthoefel et al., 2011*). Whether the intestine indeed conforms to fractal geometry has not been determined and the size-independence of the metabolic rate per cell would again seem to argue against size-dependent supply limitations. However, it is important to stress that the currently unknown putative metabolic rate scaling during the growth phase that

was discussed above leaves open the possibility of size-dependent supply limitations and thus a possible contribution of the above theories to the size-dependent energy storage in planarians.

While our results therefore do not yet provide a mechanistic explanation of why the metabolic rate of animals scales with the 3/4 power of mass, they do set the foundational basis for an experimental approach to a molecular understanding of Kleiber's law. Indeed, by demonstrating that the tremendous intra-species size fluctuations of planarians are governed by the same principle as the broader scaling law, we have established planarians as an ideal model system in which to untangle Kleiber's long-standing mystery.

# Materials and methods

## Key resources table

| Reagent type (species) or resource | Designation | Source or reference | Identifiers | Additional information |
|---|---|---|---|---|
| Strain (*Schmidtea mediterranea*) | asexual CIW4 strain of *Schmidtea mediterranea* | other | NA | obtained from Dr. Alejandro Sánchez Alvarado (Stowers Institute, Kansas City, USA) |
| Chemical compound, drug | FluoSpheres Sulfate Microspheres 4 µm, red fluorescent 580/605 nm | ThermoFisher Scientific | ThermoFisher Scientific: F8858 | See materials and methods |
| Chemical compound, drug | FluoSpheres Sulfate Microspheres 4 µm, yellow-green fluorescent 505/515 nm | ThermoFisher Scientific | ThermoFisher Scientific: F8859 | See materials and methods |
| Chemical compound, drug | LD540 lipid droplet stain | *Spandl et al., 2009* | NA | obtained from Dr. Christoph Thiele (LiMES, Universität Bonn, Germany) |
| Chemical compound, drug | Bouins fixative | TCS Biosciences | TCS Biosciences: A1602 | See materials and methods |
| Chemical compound, drug | Amyloglucosidase; AG | Sigma-Aldrich | Sigma-Aldrich: A1602 | See materials and methods |
| Chemical compound, drug | Carmine (C.I. 75470) | Carl Roth | Carl Roth: 6859.1 | See materials and methods |
| Chemical compound, drug | Richard-Allan Scientific Cytoseal XYL | ThermoFischer Scientific | ThermoFischer Scientific: 8312–4 | See materials and methods |
| Chemical compound, drug | Benzoic acid pellets; IKA C723 | IKA | IKA: 0003243000 | See materials and methods |
| Chemical compound, drug | Lipid standard: CE 16:0 D7 | Avanti Polar Lipids | Avanti Polar Lipids: 700149 | See materials and methods |
| Chemical compound, drug | Lipid standard: CholD7 | Avanti Polar Lipids | Avanti Polar Lipids: 700041 | See materials and methods |
| Chemical compound, drug | Lipid standard: TAG 50:0 D5 | Avanti Polar Lipids | Avanti Polar Lipids: 110543 | See materials and methods |
| Chemical compound, drug | Lipid standard: DAG 34:0 D5 | Avanti Polar Lipids | Avanti Polar Lipids: 110538 | See materials and methods |
| Chemical compound, drug | Lipid standard: Cer 30:1 | Avanti Polar Lipids | Avanti Polar Lipids: 860512 | See materials and methods |
| Chemical compound, drug | Lipid standard: PC 25:0 | Avanti Polar Lipids | Avanti Polar Lipids: LM-1000 | See materials and methods |
| Chemical compound, drug | Lipid standard: PE 25:0 | Avanti Polar Lipids | Avanti Polar Lipids: LM-1100 | See materials and methods |
| Chemical compound, drug | Lipid standard: PS 25:0 | Avanti Polar Lipids | Avanti Polar Lipids: 111129 | See materials and methods |

*Continued on next page*

*Continued*

| Reagent type (species) or resource | Designation | Source or reference | Identifiers | Additional information |
|---|---|---|---|---|
| Chemical compound, drug | Lipid standard: PI 25:0 | Avanti Polar Lipids | Avanti Polar Lipids: 110955 | See materials and methods |
| Chemical compound, drug | Lipid standard: SM 30:1 | Avanti Polar Lipids | Avanti Polar Lipids: 860583 | See materials and methods |
| Chemical compound, drug | Lipid standard: LPC 13:0 | Avanti Polar Lipids | Avanti Polar Lipids: 855476P | See materials and methods |
| Chemical compound, drug | Lipid standard: LPE 13:0 | Avanti Polar Lipids | Avanti Polar Lipids: 110696 | See materials and methods |
| Chemical compound, drug | Lipid standard: PG 25:0 | Avanti Polar Lipids | Avanti Polar Lipids: 111126 | See materials and methods |
| Chemical compound, drug | Lipid standard: PA 25:0 | Avanti Polar Lipids | Avanti Polar Lipids: LM-1400 | See materials and methods |
| Chemical compound, drug | Lipid standard: LPA 13:0 | Avanti Polar Lipids | Avanti Polar Lipids: LM-1700 | See materials and methods |
| Chemical compound, drug | Lipid standard: GalCer 30:1 | Avanti Polar Lipids | Avanti Polar Lipids: 860544 | See materials and methods |
| Chemical compound, drug | Lipid standard: LacCer 30:1 | Avanti Polar Lipids | Avanti Polar Lipids: 860545 | See materials and methods |
| Chemical compound, drug | Lipid standard: LPI 13:0 | Avanti Polar Lipids | Avanti Polar Lipids: 110716 | See materials and methods |
| Chemical compound, drug | Cholesteryl linoleate | Sigma-Aldrich | Cat. No.: C0289 | See materials and methods |
| Chemical compound, drug | Glyceryl trioleate | Sigma-Aldrich | Cat. No.: T7140 | See materials and methods |
| Chemical compound, drug | Linoleic acid | Sigma-Aldrich | Cat. No.: L1376 | See materials and methods |
| Chemical compound, drug | Dioleoylglycerol | Sigma-Aldrich | Cat. No.: D8894 | See materials and methods |
| Chemical compound, drug | Cholesterol | Sigma-Aldrich | Cat. No.: C8503 | See materials and methods |
| Chemical compound, drug | 1-Oleoyl-rac-glycerol | Sigma-Aldrich | Sigma, Cat. No.: M7765 | See materials and methods |
| Antibody | anti-Histone H3 | Abcam | Cat. No.: ab1791 | (1:500) |
| Antibody | anti-rabbit IRDye 680LT | LICOR | Cat. No.: 926–68023 | (1:20000) |
| Commercial assay or kit | Glucose (GO) Assay Kit | Sigma-Aldrich | Cat. No.: GAGO-20 | See materials and methods |
| Commercial assay or kit | Protein Assay Reagent | ThermoFischer Scientific | Cat. No.: 22660 | See materials and methods |
| Commercial assay or kit | Detergent Compatibility Reagent | ThermoFischer Scientific | Cat. No.: 22663 | See materials and methods |
| Other | microcalorimeter TAMIII | TA Instruments | NA | See materials and methods |
| Other | Bomb calorimeter IKA C 6000 global standards | IKA | NA | See materials and methods |
| Other | Odyssey SA Li-Cor Infrared Imaging System | LICOR | NA | See materials and methods |
| Software, algorithm (MATLAB) | MATLAB | MathWorks | NA | Algorithm to measure planarian body size available as a source file. |
| Software, algorithm | Fiji distribution of ImageJ | *Schindelin et al. (2012)* | NA | See materials and methods |

*Continued on next page*

*Continued*

| Reagent type (species) or resource | Designation | Source or reference | Identifiers | Additional information |
|---|---|---|---|---|
| Software, algorithm CellProfiler | CellProfiler (version 2.2.0 and older) | *Carpenter et al., 2006* | NA | Pipeline used for cell counting available as a source file. |

## Fitting of power laws

Power law exponents were obtained from linear fits (robust regression using a bisquare weighing function, 'robustfit' function in MATLAB) in the log-log plot. We only directly fitted the measured data. If a data set was derived from several measurements (e.g. metabolic rate vs. wet mass was derived from measurements of metabolic rate vs. dry mass and dry mass vs. wet mass), the power law estimate was computed from the original fits of the individual measurements. The respective standard error was obtained via error propagation.

## Animal husbandry

The asexual (CIW4) strain of *S. mediterranea* was kept in plastic containers in 1X Montjuïc salt water (1.6 mM NaCl, 1.0 mM CaCl$_2$, 1.0 mM MgSO$_4$, 0.1 mM MgCl$_2$,0.1 mM KCl, 1.2 mM NaHCO$_3$) with 25 mg/L gentamycin sulfate. The animals were fed homogenized organic calf liver paste and were fed at least one week prior to all experiments if not otherwise indicated. Animals were kept at 20°C before and during experiments.

## Measurement of planarian body size

Movies of gliding planarians were taken with a Nikon Multizoom AZ 100M (0.5x objective) using dark field illumination (facilitates planarian body segmentation). The following camera (DS-Fi1) settings were used: frame rate 3 Hz, exposure time 6 ms, 15 s movie length, 1280 × 960 pixel resolution, conversion factor 44 pixel/mm. Animals were placed one at a time inside a Petri dish and typically 1–4 movies taken, depending on the animal's behaviour. Movies were converted from AVI to MP4 format using Handbreak to reduce the file size. Movies were subsequently analysed using custom-made MATLAB software (MathWorks, Natick, Massachusetts, USA) (see *Figure 2—source data 2* for the MATLAB script). Typically, those frames were analysed in which the animals were gliding in a straight line (typically 10 frames). See also (*Werner et al., 2014*).

## Microcalorimetry

2–3 weeks starved size-matched planarians were placed inside 4 ml glass ampoules (TA Instruments, Cat. No.: 24.20.0401) partially filled with 2 ml of planarian water and supplemented with 10 mM HEPES for improved buffering. No HEPES was used in 22 out of 82 samples, however, no difference in animal health and/or heat generation was observed (data not shown). The ampoules were sealed with aluminium Caps (TA Instruments, Cat. No.: 86.33.0400) using a dedicated crimping tool (TA Instruments, cat. #: 3339). The measurements were performed in a multichannel microcalorimeter (TAMIII, TA Instruments), whereby 12 samples were measured simultaneously including 1–2 controls without animals. The ampoules were first inserted half way and kept in this position for 15 min in order to equilibrate with the temperature inside the device. Then, ampoules were placed completely inside the respective channels whereby they were sitting on top of a thermoelectric detector that measured the heat production in relation to an oil bath, which was kept at a constant temperature of 20°C. Before the actual measurements, the system was left to equilibrate for another 45 min. The measurements lasted between 2–3 days. Animal behaviour was not controlled and the animals were able to freely move inside the ampoule. Immediately after the metabolic rate measurements, animal dry mass was determined by drying over night at 60°C either on weighing paper or inside 0.5 ml tubes and subsequent weighing on a microbalance (RADWAG MYA 5.2Y, readability: 1 μg). The mass per animal was obtained by dividing the collective mass by the number of animals.

## Cell counting based on histone H3 protein quantification

Generating standard curves for converting Histone H3 content into cell number: cells from 15 animals (length 5–8 mm) were dissociated and counted out by FACS essentially as previously described (*Tejada-Romero et al., 2012*). Following enzymatic digestion of the tissue, the resulting cell suspension was filtered through a CellTrics 50 µm mesh (Partec, Cat. No.: 04-0042-2317) and incubated in Hoechst (33342) for 1.5 h on a rotator. Subsequently, cells were pelleted once (700 rpm, 10 min) and the supernatant replaced with fresh CMFH. The volume was adjusted to obtain a cell concentration suitable for FACS (typically 1–5x10$^6$ cells/ml). Following cell sorting, cells were kept on ice until further processing. Cells were counted with a FACS ARIA III cell sorter (Beckton Dickinson) with standard filter settings and sorted into 2 ml tubes. Typically, 10$^5$ cells were sorted per tube. Following FACS, cells were frozen at -80 °C until further use.

Determination of total cell number in different-sized planarians using quantitative Western blotting: plan area was measured using above-mentioned method (see also *Figure 2—figure supplement 1*). Subsequently, individual animals were lysed in 6 M Urea, 2% SDS, 130 mM DTT, 1 µg/ul BSA, 1 µg/ul BSA-AlexaFluor680 conjugate (ThermoFisher Scientific, Ca. No.: A34787), protease inhibitor cocktail and $\geq$ 2.5 U/ml Benzonase Nuclease (SIGMA, Cat. No.: E1014). Lysis was allowed to proceed for 1–1.5 h at room temperature, remaining tissue pieces were completely lysed by tapping the tubes and vortexing. Meanwhile, the cells for the standard curve (see above) were lysed by directly applying the lysis solution onto the frozen cells. Protein concentrations were measured in 1:5 or 1:10 dilutions using a NanoDrop spectrophotometer (Thermo Fisher Scientific) (absorbance at 280 nm). Finally, the samples were mixed with 4x Laemmli buffer (4x stock: 400 mM DTT, 200 mM Tris-HCl, 8% SDS, 40% glycerol, 0.5 mg/ml Bromophenol Blue) and incubated for 10 min at 60°C before spinning down at 13000 rpm for 5 min. The samples were run on NuPAGE Novex 4–12% Bis-Tris protein gels (Invitrogen, Cat. No.: NP0322BOX) in 1x MOPS running buffer (ThermoFisher Scientific, Cat. No.: NP0001). The loaded volumes for the standard curve corresponded to 15000, 22500, 30000, 37500 and 45000 cells (linear signal range) and the volume of the whole-animal lysates was corresponding to 50 µg of protein, ensuring that the samples were lying within the range of the standard curve. Four technical replicates were carried out per experiment (analysis of 5 individual animals) by running 2 chambers with two gels each at 140 mA for 1 h. Proteins were transferred onto Whatman Protran nitrocellulose membrane (SIGMA, Cat. No.: Z613630) for 2 h in transfer buffer (20% MeOH/1x MOPS). Membranes were blocked for 1 h at room temperature and continuous agitation in 1x TBS-T (10 mM Tris base, 150 mM NaCl, 0.1% (w/v) Tween-20, pH 7.4) and 5% (w/v) nonfat dry milk. Afterwards, membranes were incubated over night at 4°C with anti-Histone H3 antibody (Abcam, Cat. No.: ab1791) followed by at least three washes in TBS-T for 10 min. Membranes were then incubated with a fluorophore-conjugated secondary antibody (anti-rabbit IRDye 680LT, LICOR, Cat. No.: 926–68023) diluted 1:20000 in blocking solution followed by extensive washing in TBS-T (1 $\times$ 5 min, 3 $\times$ 10 min) and one final wash step in TBS (10 min). Afterwards, membranes were dried at room temperature for at least 1 h and imaged on an Odyssey SA Li-Cor Infrared Imaging System (LICOR). The relative fluorescent band intensity was quantified using the gel-analysing tool in Fiji (*Schindelin et al., 2012*). The fraction of cells from whole-animal lysates loaded onto the gel was calculated from the standard curve on each blot separately. The total number of cells in the animals was calculated as follows: number of cells loaded/volume loaded x total volume of original lysate. The obtained values were finally averaged over all four technical replicates.

## Image-based cell counting

First, plan area of individual animals was measured using above-mentioned method (see also *Figure 2—figure supplement 1*). For cell dissociation, individual animals were placed inside maceration solution (*Romero and Baguñà, 1991*) (acetic acid, glycerol, dH$_2$O at a ratio of 1:1:13 including 1 µg/ml BSA + 10 µg/ml Hoechst 33342, no methanol) and the total volume adjusted according to animal size. The solution also contained typically about 1.3 $\times$ 10$^6$ fluorescent beads/ml (FluoSpheres Sulfate Microspheres, 4 µm, red fluorescent 580/605 nm, ThermoFisher Scientific, Cat. No.: F8858) the concentration of which was determined with a Neubauer chamber for each experiment (including 10–18 animals). Dissociation was allowed to proceed at room temperature for about 15 min after which cells of remaining tissue clumps were further dissociated by taping and vortexing. Per animal, 2 µl drops of the cell suspension were pipetted into 6–10 wells of a glass bottom 96-well plate (Greiner,

Cat. No.: 655090) and the drops dried over night at room temperature. Subsequently, the entire drops were imaged on an Operetta high-content imaging system (PerkinElmer). The number of cells and beads were automatically counted using an imaging pipeline built in CellProfiler (*Carpenter et al., 2006*) (*Figure 2—figure supplement 2*). The total number of cells was calculated from each separate well/drop by the following formula: sum of cells in analysed images/sum of beads in analysed images x known total number of beads in original cell suspension. For each animal, the calculated total cell number was averaged across 9–10 wells.

## Measurement of energy content using a bomb calorimeter

Size-matched planarians (1 and 3 weeks starved) were placed inside a combustion crucible and lyophilised overnight in a lyophiliser (Heto LyoLab 3000). Then, the samples were weighed on an analytical balance (Sartorius Entris, readability: 0.1 mg) and the mass per animal was obtained by dividing the collective mass by the number of animals – thus, allowing further conversion into organismal cell numbers. Afterwards, the combustion enthalpy was measured by combustion in the presence of high pressure $O_2$ inside a bomb calorimeter (IKA C 6000 global standards) running in adiabatic mode. Benzoic acid pellets (IKA C723, Cat. No.: 0003243000) were used as a standard for calibration as well as a burning aid for the samples. In between lyophilising and combustion, the samples were kept inside a drying chamber to prevent humidification.

## Dry and wet mass measurements

To obtain the dry mass versus area and dry mass versus length scaling laws, the plan area of individual animals was measured using aforementioned method. Afterwards, animals were individually placed on round pre-weighed glass cover slips and dried over night at approximately 60°C. Subsequently, each animal was weighed 3 times on an analytical microbalance (Sartorius Research 210 P, readability: 0.01 mg) to obtain an average mass value. Wet mass was measured by removing as much of residual water as possible while individual animals were placed inside a 0.5 ml tube. After further exposing the animals to air for 30–40 min to evaporate remaining water outside of the animal, animals were weighed on a microbalance (RADWAG MYA 5.2Y, readability: 1 µg).

## Food intake assay

Plan area of individual animals (two and three weeks starved) was measured using the above-mentioned method (see also *Figure 2—figure supplement 1*). Planarians were fed with organic homogenized calf liver paste, which was mixed with about $6.5x10^5$ per 1 µl liver red fluorescence beads (FluoSpheres Sulfate Microspheres, 4 µm, fluorescent 580/605 nm, ThermoFisher Scientific, Cat. No.: F8858) coated in 1 mg/ml BSA. Single animals (or for calibration 2 µl of liver/beads mix) were dissociated into single cells in maceration solution (see above) containing 0.1% Tween-20 and about 300/µl yellow-green fluorescence beads (FluoSpheres Sulfate Microspheres, 4 µm, fluorescent 505/515 nm, ThermoFisher Scientific, Cat. No.: F8859) for volume normalization (see further below). 1 µl drops of the animal and liver macerates as well as from maceration solution only were distributed into 10 wells of a glass bottom 96-well plate (Greiner, Cat. No.: 655090) and dried over night at room temperature in the dark. Whole drops were imaged on an Operetta high content imaging system (PerkinElmer) and the number of red and yellow-green beads were automatically counted using CellProfiler (*Carpenter et al., 2006*). The volume of liver eaten per animal was calculated as follows:

- Total number of red beads per one animal = Number of red beads in 1 µl drop of worm suspension x Total volume of original maceration solution
- Total number of red beads per 1 µl liver = (Number of red beads in 1 µl drop of liver suspension/2) x Volume of maceration solution
- Volume of liver eaten per animal = Total number of red beads per one animal/Total number of red beads per 1 µl liver

To account for possible pipetting errors leading to variation in drop volumes, the volume of liver eaten per animal was normalized to the ratio between yellow-green beads in the drops of the animal macerate and in the drops of maceration solution only.

## Lipid droplet stain

Two weeks starved small worms were killed in 5% N-Acetyl-Cystein (NAC) and large worms in 7.5% NAC (5 min at room temperature) and fixed in 4% PFA for 2 days at 4°C. Fixed worms were embedded in 4% low-melting-point agarose and sectioned using a vibratome (100 μm, Leica, Germany). Sections were treated with 0.5% Triton X-100 in PBS for 2 h and incubated with lipid droplet dye LD540 (kind gift from Christoph Thiele, Bonn) (0.5 μg/ml) and DAPI (1 μg/ml) in PBS overnight at room temperature. After thoroughly washing with 0.3% Triton X-100 in PBS and a short rinse in PBS, the sections were optically cleared with the slightly modified SeeDB protocol (*Ke et al., 2013*) as follows: sections were incubated sequentially with increasing concentrations of aqueous fructose solution (25% for 4 h, 50% for 4 h, 75% and 100% fructose for overnight) and finally with the saturated fructose solution overnight. All steps were carried out at room temperature. The sections were mounted on glass slides with the SeeDB solution and confocal images were taken on a Zeiss LSM 700 inverted microscope (20x objective, Zeiss Plan-Apochromat, 0.8 numerical aperture) using 80% 2,2'-Thiodiethanol (*Staudt et al., 2007*) as immersion media.

## Lipid extraction and quantification by shotgun mass spectrometry

To assess the ability of different extraction procedures to prevent TG degradation prior to mass spectrometry, the various lipid extracts (*Figure 4—figure supplement 1A*) were analysed on high performance thin layer chromatography (HPTLC) silica gel plates (Merck, Cat.No.: 105633) using n-hexane/diethylether/acetic acid (70:30:1, vol/vol/vol) as the liquid phase (*Hildebrandt et al., 2011*). Lipids were visualized by spraying plates with 3 g cupric acetate in 100 ml of aqueous 10% phosphoric acid solution and heating at 180°C for 10 min. The following lipid standards were used for TLC: Cholesteryl linoleate (Sigma-Aldrich, Cat. No.: C0289) for Cholesterolester (CE), Glyceryl trioleate (Sigma-Aldrich, Cat. No.: T7140) for Triglyceride (TG), Linoleic acid (Sigma-Aldrich, Cat. No.: L1376) for Free fatty acids (FFA), Dioleoylglycerol (Sigma-Aldrich, Cat. No.: D8894) for Diacylglycerol (DAG); Cholesterol (Sigma-Aldrich, Cat. No.: C8503) for Cholesterol (Ch); 1-Oleoyl-rac-glycerol (Sigma-Aldrich, Cat. No.: M7765) for Monoacylglycerol (MAG).

For mass spectrometry, planarians of different size (40 small, length ~ 4 mm; 20 medium, ~ 8 mm; and 6 large, ~ 16 mm) were pooled and homogenized in ice-cold isopropanol mixed with acetonitrile (1:1). Protein amount in the homogenates was determined by BCA. 50 μg of total protein was extracted with MTBE/MeOH as described in (*Sales et al., 2017*; *Sales et al., 2016*; *Schuhmann et al., 2012*). Briefly, 700 μl of 10:3 MTBE/MeOH containing one internal standard for each lipid class was added to the dried homogenates. Synthetic lipid standards were purchased from Avanti Polar Lipids, Inc. (Alabaster, AL, USA; see key resources table). Samples were vortexed for 1 h at 4°C. Phase separation was induced by adding 140 μl of water and vortexing for 15 min at 4°C, followed by centrifugation at 13400 rpm for 15 min. The upper phase was collected, evaporated and reconstituted in 600 μl of 2:1 MeOH/CHCl$_3$. 15 μl of total lipid extract was diluted with 85 μl 4:2:1 IPA/MeOH/CHCl$_3$ containing 7.5 mM ammonium formate for mass spectrometric analysis. For the measurement of phosphatidylserines (PS), 15 μl of lipid extract were diluted with 85 ul 4:1 EtOH/CHCl$_3$ containing 0.1% triethylamine.

Mass spectrometric analysis was performed on a Q Exactive instrument (Thermo Fischer Scientific, Bremen, Germany) equipped with a robotic nanoflow ion source TriVersa NanoMate (Advion BioSciences, Ithaca, NY, USA) using nanoelectrospray chips with a diameter of 4.1 μm. The ion source was controlled by the Chipsoft 8.3.1 software (Advion BioSciences). Ionization voltage was +0.96 kV in positive and − 0.96 kV in negative mode; backpressure was set at 1.25 psi in both modes by polarity switching (*Schuhmann et al., 2012*). The temperature of the ion transfer capillary was 200°C; S-lens RF level was set to 50%. Each sample was analysed for 5.7 min. FTMS spectra were acquired within the range of m/z 400–1000 from 0 min to 1.5 min in positive and within the range of m/z 350–1000 from 4.2 min to 5.7 min in negative mode at a mass resolution of R m/z 200 = 140000, automated gain control (AGC) of $3 \times 10^6$ and with a maximal injection time of 3000 ms. Free cholesterol was quantified by parallel reaction monitoring FT MS/MS within runtime 1.51 to 4.0 min. For FT MS/MS micro scans were set to 1, isolation window to 0.8 Da, normalized collision energy to 12.5%, AGC to $5 \times 10^4$ and maximum injection time to 3000 ms. PS was measured for 1.5 min in an additional acquisition in negative FTMS mode with optimized nanoMate parameters (backpressure 1.00 psi and voltage – 2.00 kV). All acquired data was filtered by PeakStrainer (https://git.

[mpi-cbg.de/labShevchenko/PeakStrainer/wikis/home](mpi-cbg.de/labShevchenko/PeakStrainer/wikis/home)) (*Schuhmann et al., 2017*). Lipids were identified by LipidXplorer software (*Herzog et al., 2012*). Molecular Fragmentation Query Language (MFQL) queries were compiled for PC, PC O-, LPC, LPC O-, PE, PE O-, LPE, PI, LPI, PA, LPA, PS, SM, TG, DG, Cer, Chol, CE lipid classes. . The identification relied on accurately determined intact lipid masses (mass accuracy better than five ppm). Lipids were quantified by comparing the isotopically corrected abundances of their molecular ions with the abundances of internal standards of the same lipid class. The amount of lipids per animal was calculated based on the known volume of homogenization buffer and the known number of animals. Lipid amounts were normalized to cell number using the previously established scaling relationship between cell number and area (*Figure 2B*) and between length and area (*Figure 2—figure supplement 1E*).

## Histological staining for glycogen on planarian cross sections

Fixation: two weeks starved small (~4 mm) and large (13 mm −16 mm) animals were anesthetized and relaxed for 5 min on ice by supplementing chilled planarian water with 0.0485 % w/v Linalool (Sigma, L2602). Planarians were fixed in cold alcoholic Bouins fixative (15 ml Picric acid (saturated alcoholic solution, TCS Biosciences, Cat. No.: HS660), 12 ml 32% PFA, 2 ml glacial acetic acid and 15 ml ethanol) overnight at 4℃ and washed with 70% ethanol for following two days.

Paraffin embedding and sectioning: Fixed animals were dehydrated by alcohol-xylene series (1 × 10 min in 70% ethanol and 2x for 30 min in 96%, 100% ethanol and xylene, respectively). Xylene was replaced by melted paraffin at 60℃, which was exchanged three times, after 30 min, after several hours overnight and again after 30 min, which was followed by embedding. Cross-sections of 10 µm thickness were obtained using a microtome (Thermofisher Scientific, Microm HM355S). The sections were dewaxed and hydrated by xylene-ethanol series (2 × 10 min Xylene, 2 × 1 min 100%, 96% and 1 × 1 min 70%, 40%, ethanol and dH$_2$O). Prior to staining, one of the two adjacent sections was treated (for 2 h, at 37℃) with 0.2 N acetate buffer (pH 4.8) containing amyloglucosidase (0.03 U/µl) (Sigma A1602), while the other section with buffer only. By rinsing the sections with dH$_2$O, the digested glycogen was washed out on the section treated with amyloglucosidase but not on the section without enzyme treatment.

For glycogen visualization Best's Carmine staining method was used. The Carmine stock and -working solutions (Carmine (C.I. 75470) Carl Roth, 6859.1) as well as the differentiating solution were prepared as described in Romeis - Mikroskopische Technik (*Mulisch and Welsch, 2010*). The sections were treated for 10 min with Carmine working solutions following by differentiating solution 2x for 1 min. Sections were briefly rinsed with 80% ethanol and treated 2x for 1 min with 100% ethanol and 2x for 2 min with xylene and mounted in CytosealXYL (Richard-Allan Scientific; 8312–4). Stained sections were imaged with an Olympus BX61 Upright Microscope with 5x and 20x objectives.

## Glycogen assay

Two weeks starved animals were homogenized in dH$_2$O (40 worms of 4 mm length in 0.5 ml, 20 worms of 8 mm in 1 ml and 10 worms of 16 mm in 1 ml) using zirconia/silica beads (1.0 mm diameter, Carl Roth GmbH + Co. KG, Cat.No:11079110z) at 4℃ for 10 min. After brief centrifugation, the samples were flash frozen in liquid nitrogen and sonicated (Covaris S2 Sonicator) for 1 min. The homogenate was used for glycogen and total carbohydrate quantifications. The glycogen quantification method was adapted to planarians based on a protocol for *Drosophila* larvae from the C. Thummel lab (University of Utah). Heat-treated homogenate (70℃, 10 min) was centrifuged at 13400 rpm for 2 min and the supernatant was taken for the measurements. The extracted glycogen was digested to glucose by amyloglucosidase treatment (Sigma, Cat. No.: A1602) (0.015 U/µl of 0.2 M acetate buffer, pH 4.8) for 2 h at 37℃. The glucose content was measured using the glucose assay kit (Sigma, Cat. No.: GAGO-20). The assay was performed in black glass bottom 96-well plates (Greiner Bio-One, Cat. No.: 655090) and the absorption spectra was measured using Envision Microplate Reader (Perkin Elmer). Additionally, to assess background levels of free glucose, the supernatant without amyloglucosidase treatment was measured. Planarians do not contain free glucose at detectable levels (data not shown). Glucose and glycogen amounts were determined using a standard curve built on a glucose and glycogen dilution series, respectively. Glycogen extraction using hot 30% KOH (*Figure 4—figure supplement 1D*) was performed as previously published (*Rasouli et al., 2015*).

## Total carbohydrate measurement

Determination of total carbohydrate was carried out on whole homogenates (same as used in glycogen assay) using the phenol-sulfuric acid method. In brief, the homogenate was heated with the 96% $H_2SO_4$ at 90°C for 15 min, mixed with phenol (saturated with 0.1 M citrate buffer, pH 4.3, Sigma, Cat. No.: P4682) (Homogenate: $H_2SO_4$: phenol at a ratio of 1:5:1) and subsequently distributed into a 96-well plate (Thermo Scientific Nunc, Cat. No: 167008). The absorbance was measured at 492 nm Envision Microplate Reader (Perkin Elmer). Carbohydrate amounts were determined using a standard curve built on a glycogen dilution series. The amount of glycogen and total carbohydrates per animal was calculated based on the known volume of homogenisation buffer and the known number of animals. Glycogen and carbohydrate amounts were normalised to organismal cell number using the previously established scaling relationship between cell number and area (*Figure 2B*) and between length and area (*Figure 2—figure supplement 1E*). The non-glycogen carbohydrate amount was calculated by subtracting the determined glycogen from the carbohydrate amount.

## Protein measurements

Planarians of approximately 4, 8 and 16 mm length were chosen and protein amounts were determined using the Pierce 660 nm Protein Assay Reagent (ThermoFisher Scientific, Cat. No.: 22660) according to the manufacturer's instructions. To ensure compatibility with the used lysis solution (see below), the Pierce 660 nm Protein Assay Reagent was complemented with Ionic Detergent Compatibility Reagent (ThermoFisher Scientific, Cat. No.: 22663). Planarian lysates were prepared as follows: 44 small (length 4 mm), 10 medium (8 mm) and 10 large (16 mm) animals were placed inside 1.5 ml tubes and rinsed once with $dH_2O$. A lysis solution containing 10 M Urea, 2% SDS, 130 mM DTT, 2.5 µg/ml Benzonase (home-made) and a protease inhibitor cocktail was added and the animals incubated for 10 min followed by homogenisation using a motorized plastic pestle. Volumes of lysis buffer used were 235 µl for small, 335 µl for medium and 2 ml for large animals. Subsequently, lysates were cleared by centrifugation at 13000 rpm for 1 min. The assay was performed in black glass bottom 96-well glass bottom plates (Greiner Bio-One, Cat. No.: 655090) and the resulting absorption spectra measured using a FLUOstar Omega Microplate Reader (BMG LABTECH).

## Whole mount in situ hybridization

Whole mount in situ hybridization (WISH) was essentially performed as previously described (*King and Newmark, 2013*; *Pearson et al., 2009*).

## Statistics

All statistical analyses were carried out using GraphPad Prism version 7.0 c for Mac OSX (GraphPad Software, La Jolla, California, USA).

## Software

Excel for Mac (Microsoft, Redmond, Washington, USA) and KNIME (*Berthold et al., 2007*) (KNIME AG, Zurich, Switzerland) were used for data handling and calculations; GraphPad Prism v7.0c (GraphPad Software, La Jolla, USA) was used for statistical analyses and data visualization; MATLAB (MathWorks, Natick, Massachusetts, USA) was used for planarian body size measurements, theoretical analysis of models, data handling and visualization; CellProfiler (*Carpenter et al., 2006*) was used for image analysis; Fiji (*Schindelin et al., 2012*) was used for Western blot quantification and image processing; Adobe Photoshop CS5 and Illustrator CS5 (Adobe Systems, San Jose, California, USA) were used for image processing and generating figures; the manuscript was prepared for submission using Word for Mac (Microsoft, Redmond, Washington, USA) or MATLAB (MathWorks, Natick, Massachusetts, USA).

## Data and materials availability

All data on which the conclusions of this paper are based are presented in the figures, figure supplements or source data that was submitted with this manuscript.

## Acknowledgements

We thank N. Alt, J. Richter, J. Ferria, A. Mishra, M. Toth and I. Smith for size quantifications, K-O Linde and IKA Werke GmbH and Co. for microcalorimetry support, Dr. Passant Atallah (IPF Dresden) and Dr. Michał Surma (Lipotype GmbH) for support with the animal mass measurements and H. Andreas and S. von Kannen for technical support. We thank S. Weiche from the CMCB technology platform TU Dresden (EM and Histology) and the following MPI-CBG core facilities for their support: Cell technologies, Technology development studio, and scientific computing. We thank Dr. Carl Modes for discussions and comments on the manuscript.

## Additional information

### Competing interests

Frank Jülicher: Reviewing editor, *eLife*. The other authors declare that no competing interests exist.

### Funding

| Funder | Grant reference number | Author |
|---|---|---|
| Max-Planck-Gesellschaft | Open-access funding | Albert Thommen<br>Olga Frank<br>Oskar Knittelfelder<br>Andrej Shevchenko<br>Frank Jülicher<br>Jochen C Rink |
| Bundesministerium für Bildung und Forschung | 031 A 099 | Steffen Werner<br>Benjamin M. Friedrich |

The funders had no role in study design, data collection and interpretation, or the decision to submit the work for publication.

### Author contributions

Albert Thommen, Olga Frank, Conceptualization, Data curation, Formal analysis, Investigation, Visualization, Methodology, Writing—original draft, Writing—review and editing; Steffen Werner, Conceptualization, Data curation, Software, Formal analysis, Investigation, Visualization, Methodology, Writing—original draft, Writing—review and editing; Jenny Philipp, Investigation, Writing—review and editing; Oskar Knittelfelder, Data curation, Formal analysis, Investigation, Writing—review and editing; Yihui Quek, Investigation, Methodology; Karim Fahmy, Resources, Supervision, Funding acquisition, Writing—review and editing; Andrej Shevchenko, Resources, Formal analysis, Supervision, Funding acquisition, Writing—review and editing; Benjamin M Friedrich, Conceptualization, Supervision, Funding acquisition, Writing—review and editing; Frank Jülicher, Conceptualization, Resources, Supervision, Funding acquisition, Writing—review and editing; Jochen C Rink, Conceptualization, Resources, Supervision, Funding acquisition, Writing—original draft, Writing—review and editing

### Author ORCIDs

Albert Thommen (iD) http://orcid.org/0000-0002-2285-0714
Steffen Werner (iD) http://orcid.org/0000-0002-8764-1366
Olga Frank (iD) http://orcid.org/0000-0003-2345-3136
Oskar Knittelfelder (iD) http://orcid.org/0000-0002-1565-7238
Yihui Quek (iD) http://orcid.org/0000-0002-1227-0804
Karim Fahmy (iD) http://orcid.org/0000-0002-8752-5824
Andrej Shevchenko (iD) http://orcid.org/0000-0002-5079-1109
Benjamin M Friedrich (iD) http://orcid.org/0000-0002-9742-6555
Frank Jülicher (iD) https://orcid.org/0000-0003-4731-9185
Jochen C Rink (iD) http://orcid.org/0000-0001-6381-6742

## Decision letter and Author response
Decision letter https://doi.org/10.7554/eLife.38187.034
Author response https://doi.org/10.7554/eLife.38187.035

## Additional files

### Supplementary files
• Supplementary file 1. List of scaling relationships.
DOI: https://doi.org/10.7554/eLife.38187.029

• Transparent reporting form
DOI: https://doi.org/10.7554/eLife.38187.030

### Data availability
All data generated or analysed during this study are included in the manuscript and source data files. Source data files have been provided for all main figures 1-5 and figure 1 - figure supplement 1, figure 2 - figure supplement2 and figure 3 - figure supplement 3.

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

# Appendix 1

DOI: https://doi.org/10.7554/eLife.38187.031

## Implementation of the theoretical model

The model describes the dynamic changes of total physiological energy $E$, defined as the fraction of energy in the body that can be metabolized and released as heat, see **Figure 3A**. The physiological energy $E$ thus decreases due to metabolic heat production $P$ and increases due to feeding, where $J$ captures the net influx of physiological energy (taking into account a potentially elevated metabolism during feeding): $\dot{E} = J - P$. The dot denotes the time derivative. The average energy per cell is computed by dividing the total physiological energy by the total number of cells $N$: $e = E/N$. Thus, the energy per cell changes according to: $\dot{e} = j - p - Ke$, where we define $j = J/N$, $p = P/N$ and the growth rate $K = \dot{N}/N$. An increasing cell number decreases the energy per cell.

Paradigm 1 assumes that cell division and cell death directly depend on the energy available per cell $e$. For simplicity, we consider a linear relationship between the growth rate and the energy per cell: $K = K_0(e/e_s - 1)$, with $K_0$ being a characteristic rate of growth and degrowth and $e_s$ being the critical energy per cell at which planarians switch between growth and degrowth (**Figure 3—figure supplement 1A**, dashed line). Thus, we can describe the energy dynamics by $\dot{e} = j - p - K_0(e/e_s - 1)e$. During starvation, $j = 0$ and the growth rate is decreasing, which requires $\dot{e} < 0$ (red curve). The maximum of $\dot{e}$ is at $e = e_s/2$ and from $\dot{e} < 0$ follows that $p > e_s K_0/4$. During feeding, where $j > p$ the curve is shifted upwards and $e$ ends up in a growth regime (blue curve). For a constant energy influx $j$, the equation for $e$ has a stable fixed point $e^* = \frac{e_s}{2} + \sqrt{\left(\frac{e_s}{2}\right)^2 + \frac{(j-p)e_s}{K_0}}$ with $e^* > e_s$ for $j > 0$. Thus, the animal would grow at a constant rate $K^* = K_0(e^*/e_s - 1)$. In order for the growth rate to decrease with animal size (**Figure 2E**), the energy influx per cell $j(N)$ must not be constant but has to be a decreasing function of $N$, hence, we choose $j(N) = j_0/(1 + N)$.

**Figure 3—figure supplement 1B** shows a time course of the organismal cell number $N$ when going through several rounds of feeding (blue) and starvation (red), always switching at a certain size, specifically at $N = 0.5 \times 10^6$ cells and $N = 4.5 \times 10^6$ cells (lower and upper dashed lines, respectively). In the beginning of the starvation interval, we see an overshoot where the animal still grows although feeding has stopped. As a result, we observe rather generic growth and degrowth kinetics, independent of initial values for energy and cell number or the feeding scheme, see **Figure 3—figure supplement 1C**. Any perturbation decays quickly and there is no strong dependence on feeding history.

Paradigm 2 and 3 assume a constant relationship between cell number and physiological energy content of the worm: $E = \alpha N$ and $E = \beta N^{c+1}$, respectively, with proportionality constants $\alpha$ and $\beta$. In consequence $\dot{N}/N = \dot{E}/E$ and $\dot{N}/N = \dot{E}/(E(c+1))$, respectively, which can be related to metabolic rate and energy influx from feeding via $\dot{E}/E = -P/E$ during degrowth and via $\dot{E}/E = (J - P)/E$ during growth. In paradigm 2, $E/N$ is constant, therefore both $P/N$ and $J/N$ have to depend on $N$ to explain the size-dependence of the growth and degrowth rates. In paradigm 3, both $J/N = \delta_1$ and $P/N = \delta_2$ can be chosen to be constant. In consequence, we obtain $\dot{N} = \delta N^{1-c}$ with $\delta = (\delta_1 - \delta_2)/\beta/(c+1)$ and finally a power law for the growth/degrowth dynamics: $N(t) = (\delta c t + N(0)^c)^{1/c}$.

To fit the growth dynamics in **Figure 2C** by paradigm 1, we use the following parameters: $j_0/e_s = 14\%/d$, $p/e_s = 1.3\%/d$, $K_0 = 4.3\%/d$, initial conditions $N(0) = 3 \times 10^6$ and $e(0)/e_s = 1$ as well as a switch between feeding and starvation regimes at $N = 0.05 \times 10^6$ and $N = 6.5 \times 10^6$. Yet, several combinations of parameter values can fit the measurement equally well. From a fit of paradigm 2 in **Figure 2C**, we obtain $P/E = 165 \ N^{-0.35}\%/d$ and $J/E = 155 \ N^{-0.25}\%/d$. Finally, from a fit of paradigm 3 to the data, we obtain $E/P = 0.45 \ N^{0.35}d$ and $J/P = 3.7$, see **Figure 2C**. Independent fits with different exponents $c$ for growth and degrowth dynamics (corresponding to non-constant ratio $J/P$) do not yield a better agreement with the data than our minimal model.

## Appendix 2

DOI: https://doi.org/10.7554/eLife.38187.031

### Prediction of organismal cell number from Kleiber's law

The size-independent value of the metabolic rate/cell in *S. mediterranea* that we measure here is about 1 pW (*Figure 3D*). As noted in the text, this is very similar to reported values of the metabolic rate/cell in humans (1–5 pW; *Bianconi et al., 2013*; *Purves and Sadava, 2004*). If we assume an approximately constant and mass-independent value of the metabolic rate/cell of 1 pW across the whole animal kingdom and solely mass allometries as the basis of the 3/4 exponent, Kleiber's law should be able to accurately predict the total organismal cell numbers of all animals. This is because the total metabolic rate $P$ of the organism (given by Kleiber's law) is defined as the cellular metabolic rate multiplied by the total number of cells. Thus, $P/$(1pW) can provide a first order estimate for the cell number of each animal. For example, *Figure 1E* would yield a total metabolic rate of 60–200 W (in agreement with *Purves and Sadava, 2004*) for a human of 70 kg, amounting to $6–20 \times 10^{13}$ cells. This is in close agreement with the current estimate of $3.7 \times 10^{13}$ cells (*Bianconi et al., 2013*) and therefore suggests that an allometric scaling relationship of cell number (rather than cellular metabolic rate) with body mass is a plausible general cause of Kleiber's law.

