## [Decision Letter]

Thank you for submitting your article "Body size-dependent energy storage causes Kleiber's law scaling of the metabolic rate in planarians" for consideration by *eLife*. Your article has been reviewed by Naama Barkai as the Senior Editor, a Reviewing Editor, and three reviewers. The following individuals involved in review of your submission have agreed to reveal their identity: Iswar K Hariharan (Reviewer #1); Alejandro Sánchez Alvarado (Reviewer #3).

The reviewers have discussed the reviews with one another and the Reviewing Editor has drafted this decision to help you prepare a revised submission.

As you will see, the reviewers enjoyed your study, finding it novel and well executed. Reviewer #3, however, raised some doubt about your model, which we kindly like you to consider in full, and potentially modify the model/ story accordingly. The reviewers agreed that the other comments can be addressed by discussion. The additional experiments that were suggested are not essential, although they can improve the story, and we therefore leave it to you to decide whether to add them or not.

Reviewer #1:

This is an interesting an unusual manuscript that explores mechanisms underlying Kleiber's law which states that when different organisms are compared, the organismal metabolic rate is proportional to 3/4 power of the body mass. The authors of this study have chosen to explore this law within a single species, the planarian *S. mediterranea* which can have body sizes that vary over a 40-fold range. Remarkably, they find that this law does indeed hold over this range. Moreover, the authors find that the average metabolic rate per cell is size-independent. However, the energy content per cell increases with organismal size likely because of increased glycogen and triglycerides in the cells of larger animals. Thus, as mass increases, the cell number increases only to the 0.74 power of mass with individual cells presumably weighing more to account for this difference.

There are a lot of careful measurements and a lot of interesting data. This manuscript makes an important contribution to our understanding of organismal size regulation and I am enthusiastic about this study. I would like the authors to address the following points prior to publication.

1) I did find portions of the manuscript difficult to understand without delving into legends to figure supplements or the methods section. I suspect that most of the readers of this paper will not be working on size regulation in planaria. I would prefer if the authors explained their experiments more clearly even at the expense of making the paper longer. For example, I had erroneously assumed that the measurements of metabolic rate were based on individual worms until I saw the cartoon in Figure 1—Figure supplement 1 and then realized that the authors use multiple worms of similar size (how many?).

2) While the discoveries that authors have made might explain what happens in planaria of different sizes, I am not convinced that this explains Kleiber's law as it pertains to different species. If this was a general principle, then one would expect that the cells mass (or organismal energy stores) correlates with organism size between species. Is there any evidence for or against this? There is a single sentence in the discussion that alludes to this. However, since the authors have framed the problem in the context of interspecific comparisons, there should be more discussion about whether this is an intraspecific phenomenon or whether it could also account for interspecific differences.

3) If a large planarian is cut into two equal pieces, the prediction from this study is that each half would have an excess of storage for its new size and would presumably readjust. Is this what the authors think will happen? I think this would be a nice testable prediction of this model. I am simply suggesting this experiment – not asking for it as a requirement for publication.

4) I am a bit confused by Figure 2 C and D. The Y-axis in these panels is on a log scale. Are the Y-axes of the insets on a linear scale (indicative of exponential growth and degrowth)? If so, this should be stated in the legend or shown in the figure.

Reviewer #2:

The power law relationship between body metabolic rate and body mass has garnered nearly 100 years of correlative observations and a wealth of theoretical approaches for understanding. Yet, it remains unclear in any organism how this fundamental scaling property can be explained in terms of developmental mechanisms. Planarians undergo reversible scaling over a large range as adults and this study makes use of them to uncover the strategy by which these animals obey the 3/4 power relationship Kleiber's law, determining that increases to body size cause allocation of more resources toward energy storage and away from growth.

First the authors measure metabolic rate using calorimetry in order to determine heat expenditure of live planarians across a range of sizes to find that metabolic rate scales with a 3/4 power relationship compared to mass. To investigate the relationship between metabolic rate and growth rates, planarians of different sizes and feeding histories were examined for changes to overall size (measured as area of moving animals) and related to overall cell numbers through Histone H3 protein abundance or direct counting of macerated or H3P-labeled cells. Consistent with a model in which large animals have reduced metabolic rate, large animals have lower rates of growth. Surprisingly though, large animals also undergo lower rates of degrowth and these relationships did not depend on feeding history but rather only on size. In order to gain insight into the systems-level mechanisms that could account for lower metabolic rates in larger animals, the authors considering three potential models relating per cell energy intake, content, and expenditure of energy with respect to size. To distinguish these, further experiments set to measure energy intake (J) as volume of liver paste consumed, overall energy (E) content from bomb calorimetry, and overall energy loss (P) as heat expenditure in live animals as before. Energy loss normalized to total cell number did not vary across animals of different size, and food consumption normalized to cell number also did not consistently vary. By contrast, total animal energy normalized to total cell number increased with animal size, providing a candidate systems level explanation for Kleiber's law. The authors tested the hypothesis that larger animals may divert more energy into producing energy storage molecules and find that large animals contain substantially greater relative abundance of triglycerides and glycerol produced within or nearby the planarian gastrovascular system (intestine). Therefore, a shift toward energy storage as size increases results in lower energy average expenditure per cell or mass unit. Based on prior theoretical work in generalized systems, the source of the particular 3/4 power relationship could then arise from intrinsic geometric limitations relating 2-dimensional branched structures involved in energy uptake (such as the planarian intestine) providing energy to surrounding 3-dimensional tissue.

Planarians are an excellent model for studying growth and this study represents a major step forward for understanding a fundamental property relating size to growth rate. The work is highly original, and the quality of the data and analysis is superb. Additionally, this study provides a wealth of new information about planarian growth/degrowth and establishment of assays to facilitate subsequent studies in this system. By offering a new explanation for a fundamental property, the work is of high impact. I have comments below to further clarify the findings.

1) It is somewhat unclear why these three particular theoretical models were assessed as potential explanations for the relationships between energy consumption, storage and expenditure. Do they represent the scope of possible mechanisms accounting for energy content with respect to cell number? Also, the way that the presentations of theory and experiment flow in the paper is somewhat awkward. In particular, experimental observations of growth/degrowth rates as independent of feeding history are preceded by consideration of a model (model 1) that I believe assumes otherwise.

2) Energy influx is measured here as volume of homogenized calf liver consumed, in turn measured as a proxy of spiked in microbeads consumed. However, it seems possible that the per volume amount of nutrient absorption from feeding could depend on animal size. Perhaps a comparison between undigested and digested/defecated calf liver could be used to provide a more direct measure of J.

3) It is somewhat surprising that total protein only increases ~20% in animals that differ 4x in size, given prior information about several specific cell types (brain cells, eye cells, etc.) whose numbers scale with total animal size. Are particular or most cells from large animals smaller than those of small animals? Alternatively, could the explanation be in differences of cell type allocation between small and large animals?

4) After substantial tissue removal through surgery, fragments from large planarians regenerate into small planarians, probably due to an inability to consume food during regeneration. Most fragments would then initiate regeneration with a storehouse of energy molecules within the intestine. What happens to glycogen or triglyceride storage during this rapid attainment of smaller size?

5) If overall growth rate in terms of animal area is lower for larger animals, what is a potential explanation for the linear relationship between mass and cell number that also includes numbers of proliferative cells (labeled by H3P)?

*Reviewer #3:*

This very interesting article aims to address how scale and proportion may be regulated in an animal with marked developmental plasticity like the planarian *Schmidtea mediterranea*. The authors put forward the very bold claim that during planarian growth and degrowth, Kleiber's law does not emerge from a size-dependent decrease in cellular metabolic rate, but from a size-dependent increase in mass per cell. We find this conclusion both intriguing and potentially illuminating in addressing efforts to mechanistically dissect the regulation of scale and proportion in living organisms.

There are, however, a number of issues we would like to have the authors clarify and a number of questions we would like them to address.

1) Why was body plan used as the main measure in these studies? Given that different measurements of animal size (e.g., number of cells, wet and dry weight, length, body plan) are not necessarily interrelated, it is not clear to us why this particular metric was selected.

2) Along the above lines, why not use animal volume instead as a measurement of size? This could be determined by immersing an animal into a tube filled with a defined amount of water (or other liquid) and comparing the water level change before immersion and after immersion, for example.

3) The reported dependence of the degrowth rate on cell number (animal size) is counterintuitive. The claim is made that large animals lose mass (volume) very slowly, while at the same time small animals display fast degrowth. If we understand the premise put forward, this would mean that small animals should disappear immediately given the high rate of degrowth. Moreover, it is also clear that the cellular content of the animal with a given size may affect the growth/degrowth rate. As such, it is difficult to follow the mathematical logic put forward to explain the recorded observations.

4) We have a few concerns regarding the energy balance models. We are including in a separate document (link below) a test of the mathematical models/assumptions. These were tested because we are having difficulty understanding the authors' proposal that the entirety of animal growth rate depends on energy per cell. Assuming a uniform distribution of animal energy over all cells is not justified - there might be strong fluctuations in the energy of individual cells (cephalic ganglia, parenchymal cells, and other organs, for example). Please consult our tests of Paradigms 1 and 3. We would appreciate hearing from the authors to help us understand the noted discrepancies.

Link to document: https://submit.elifesciences.org/elife_files/2018/05/28/00052911/00/52911_0_attach_24_16058.pdf

---

## [Author Response]

Reviewer #1:1) I did find portions of the manuscript difficult to understand without delving into legends to figure supplements or the methods section. I suspect that most of the readers of this paper will not be working on size regulation in planaria. I would prefer if the authors explained their experiments more clearly even at the expense of making the paper longer. For example, I had erroneously assumed that the measurements of metabolic rate were based on individual worms until I saw the cartoon in Figure 1—Figure supplement 1 and then realized that the authors use multiple worms of similar size (how many?).

Done. We have expanded the text of multiple manuscript sections in order to make the technical underpinnings of the experiments more accessible. For example, the text section pertaining to Figure 1 now reads:

**“**The size-dependence of the metabolic rate was measured by enclosing cohorts of size-matched and two to three weeks starved animals in vials and measuring their heat emission over a period of > 24 h (Figure 1—figure supplement 1). Animal numbers/vial varied between 2 (= largest size cohort) and 130 (= smallest size cohort) in order to yield measurements with comparable signal-to-noise ratios”.

2) While the discoveries that authors have made might explain what happens in planaria of different sizes, I am not convinced that this explains Kleiber's law as it pertains to different species. If this was a general principle, then one would expect that the cells mass (or organismal energy stores) correlates with organism size between species. Is there any evidence for or against this? There is a single sentence in the discussion that alludes to this. However, since the authors have framed the problem in the context of interspecific comparisons, there should be more discussion about whether this is an intraspecific phenomenon or whether it could also account for interspecific differences.

We agree. The revised Discussion section now includes an entire paragraph on this subject, which reads:

“Size-dependent energy storage as the physiological basis of Kleiber’s law in planarians raises the question whether the same principle might also apply in other animals. […] Moreover, the hypothesis that mass allometries rather than metabolic rate allometries generally account for Kleiber’s law might be interesting to explore in other species.”

3) If a large planarian is cut into two equal pieces, the prediction from this study is that each half would have an excess of storage for its new size and would presumably readjust. Is this what the authors think will happen? I think this would be a nice testable prediction of this model. I am simply suggesting this experiment – not asking for it as a requirement for publication.

Thanks, this is an intriguing experiment that has been high on our experimental to-do list since a while. Indeed, our findings predict that the energy stores of tissue pieces should reduce to the level appropriate for the new size. However, a compounding problem is that cutting also activates regeneration, which we know to significantly increase the metabolic rate above resting levels (e.g., Allen, 1919; Osuma et al., 2018). The increased energetic demands of regeneration have to be powered by tissue intrinsic energy reserves, since regeneration can be completed without feeding. Hence regenerative processes also deplete metabolic energy stores and teasing apart the relative contributions of wound-induced from size-dependent effects becomes tricky

A simple way to disentangle the relative contributions would be to carry out the measurements in animals with blocked regeneration. However, both stem cell depletion by irradiation or follistatin/activin (RNAi) as standard “regeneration blockers” significantly alter whole animal physiology and the results would therefore be of questionable utility. We have therefore decided to take the long route and have initiated the establishment of O_2_ consumption measurements in order to be able to quantify the acute, wound-induced increase in the metabolic rate (which is not possible with our current microcalorimetry set-up). Further, we are dissecting the metabolic pathways and energy stores that power regeneration. The hope is that we will be able to quantitatively measure and specifically interfere with the wound-induced metabolic component and thus to have a definitive understanding of the size effect. Collectively, these efforts constitute a significant effort involving multiple projects in the lab and we hope that the reviewer agrees with our decision to save the preliminary data in this respect for a separate publication.

4) I am a bit confused by Figure 2 C and D. The Y-axes in these panels is on a log scale. Are the Y-axes of the insets on a linear scale (indicative of exponential growth and degrowth)? If so, this should be stated in the legend or shown in the figure.

Done. Indeed, the Y-axes of the main panels in Figure 2C and D are on a log scale, due to the power law nature of the growth process. Note that the curves are not perfectly straight and parallel, thus indicating deviations from exponential growth. On short time scales, the curves in Figure 2C-D can nevertheless be approximated by exponentials. In fact, this is used to determine growth rates for each size regime in Figure 2E.

Note that we removed the insets in the revised version of the manuscript in response to comments of reviewer 3 (see below).

Reviewer #2:1) It is somewhat unclear why these three particular theoretical models were assessed as potential explanations for the relationships between energy consumption, storage and expenditure. Do they represent the scope of possible mechanisms accounting for energy content with respect to cell number?

Indeed, the three models are non-exhaustive as more complex or mixed scenarios are possible. However, the models cover the three principal possibilities of the organismal energy content *E* being a dynamic, size-invariant or size-dependent variable. To make this clear in the text, we have inserted the following sentence:

“Although more complex scenarios are possible, the three paradigms cover the three principal possibilities of *E* as dynamic (paradigm 1), size-invariant (paradigm 2) or size-dependent variable (paradigm 3).”

Also, the way that the presentations of theory and experiment flow in the paper is somewhat awkward. In particular, experimental observations of growth/degrowth rates as independent of feeding history are preceded by consideration of a model (model 1) that I believe assumes otherwise.

Thanks for pointing out this incongruency in the text. The inference of history-independence on basis of the growth/degrowth measurements is not entirely waterproof to the measurement noise and a history-dependence of energy stores therefore remained a possibility worth exploring. To better emphasize the tentative nature of the history independence, the relevant text section now reads:

“Interestingly, the degrowth rates appeared to be independent of feeding history and thus primarily a function of size (Figure 2—figure supplement 3B)”.

And, when discussing the experimental evaluation of the models:

“The experimentally measured gross energy content and the physiologically accessible energy content *E* (green and black solid lines in Figure 3F) differ by a constant factor of about two, consistent with the previously inferred size- rather than feeding history dependence of the organismal growth rate (Figure 2 —figure supplement 3B).”

2) Energy influx is measured here as volume of homogenized calf liver consumed, in turn measured as a proxy of spiked in microbeads consumed. However, it seems possible that the per volume amount of nutrient absorption from feeding could depend on animal size. Perhaps a comparison between undigested and digested/defecated calf liver could be used to provide a more direct measure of J.

We agree that the ingested food volume is a rather indirect measure of *J*. Unfortunately, the suggested quantification of fecal mass is challenging due to such factors as post-excretion decomposition by bacteria, partial solubilization in the culture supernatant and other sources of mass deposition in the culture dish, e.g. the significant mucus secretion. Instead, we have initiated the quantification of protein absorption coefficients by means of feeding heavy isotope labeled liver and subsequent quantification of the planarian proteome. Although the preliminary results are consistent with a size-independence of *J*, they unfortunately raised a number of technical caveats that will require further significant experimental investments to sort out conclusively. Our finding that the experimentally measured gross energy content and the modeled physiologically accessible energy content E (green and black solid lines in Figure 3F) differ by a constant factor of about 2 provides at least an indirect indication that food assimilation does not change substantially between small and large animals.

We therefore hope that the reviewer agrees with our decision to include the SILAC data in a future manuscript.

3) It is somewhat surprising that total protein only increases ~20% in animals that differ 4x in size, given prior information about several specific cell types (brain cells, eye cells, etc.) whose numbers scale with total animal size. Are particular or most cells from large animals smaller than those of small animals? Alternatively, could the explanation be in differences of cell type allocation between small and large animals?

We agree that these data raise a number of important and interesting questions. First, please, note that the mass constituents in Figure 5C were normalized to total cell number. Hence it is the amount of protein per cell that increases by 23% whereas the absolute amount of protein per animal would actually increase by 2205% (23-fold) when comparing a 16 mm vs. 4 mm long animal. Our data currently cannot single out the exact source of the extra protein/cell. Possibilities include i) size-dependent changes in general cell size (i.e., larger cell size in large animals), ii) general size-invariance of most cell types, but major size-dependent changes in the size of a specific cell type (e.g., intestinal phagocytes), iii) allometric organ scaling and thus size dependent changes in cell type composition (e.g., more protein-rich secretory cells in larger animals) or iv) no changes at the cell level, but a size-dependent increase in the amount of extracellular matrix deposition. We are addressing all possibilities as part of our long-term effort to understand size scaling in planarians. However, at present, we cannot quantitatively explain the above effect and therefore opt to not include further data.

4) After substantial tissue removal through surgery, fragments from large planarians regenerate into small planarians, probably due to an inability to consume food during regeneration. Most fragments would then initiate regeneration with a storehouse of energy molecules within the intestine. What happens to glycogen or triglyceride storage during this rapid attainment of smaller size?

Indeed, this is an intriguing experiment. Please see our detailed response to point 3 of reviewer 1, who suggested the same experiment.

5) If overall growth rate in terms of animal area is lower for larger animals, what is a potential explanation for the linear relationship between mass and cell number that also includes numbers of proliferative cells (labeled by H3P)?

This point may reflect an underlying misunderstanding- please note that we do not see a linear relationship between mass and cell number: Figure 5A is a log-log plot and thus illustrates an exponential power-law relationship between cell number and mass (caused by increased energy storage). Assuming that the reviewer refers to the puzzling decrease of the growth rate in large animals in the face of the size-independence of metabolic rate, we have added a discussion paragraph on this subject. Specifically, we stress the fact that all measurements in this manuscript have been done on 2-3 weeks starved animals, while growth mainly occurs within ~5 days of feeding. This leaves open the possibility of a size-dependence of the metabolic rate during the feeding response and an associated decrease in the fraction of the proliferating neoblasts as ultimate explanation of the size-dependence of the growth rate.

An experimental verification of this hypothesis will require the establishment of assays to measure acute changes in the metabolic rate (which is not possible with our current microcalorimetry set-up), but also a battery of experiments on the size scaling of the stem cell fraction. All are currently in progress as part of our long-term effort to understand planarian growth and degrowth dynamics, but we hope that the reviewer agrees that the respective data would go beyond the scope of the current manuscript.

Reviewer #3:There are, however, a number of issues we would like to have the authors clarify and a number of questions we would like them to address.1) Why was body plan used as the main measure in these studies? Given that different measurements of animal size (e.g., number of cells, wet and dry weight, length, body plan) are not necessarily interrelated, it is not clear to us why this particular metric was selected.

Indeed,“size” is a colloquial term that encompasses mass, cell number, length, width, plan area, volume, cell number and other metrics. Although specific size metrics may indeed not interrelate linearly, they are nevertheless systematically interdependent. We explicitly demonstrate this point in the form of the various scaling laws that we measure and report in the manuscript. The practical significance of the scaling laws is that we can infer a metric of interest indirectly from measurements of a different metric. As justified in the text, our choice of plan area as principal size metric is motivated by the pragmatic reason that it is a simple, accurate and non-invasive metric that we can repeatedly acquire with minimal disturbances to the animal (see text and supplementary material for supporting data). Volume or wet weight quantifications would have been similarly useful for the purposes of this manuscript but turned out to be more cumbersome and less accurate in practice (see below).

2) Along the above lines, why not use animal volume instead as a measurement of size? This could be determined by immersing an animal into a tube filled with a defined amount of water (or other liquid) and comparing the water level change before immersion and after immersion, for example.

No doubt, volume measurements would make a useful metric for understanding planarian scaling. Though the suggested method might work in principle, the capillary forces along the tube walls will render volume quantifications of smaller worms rather inaccurate (especially in the case of small animals with only a few µl of total volume). Instead, we have evaluated commercial laser scanners to obtain 3D worm surfaces as volume approximation. Although principally feasible, the swift movement of the worms in conjunction with the data acquisition rates proved challenging in practice. Nevertheless, we will keep on tryingand hopefully have a volume assay for the next manuscript.

3) The reported dependence of the degrowth rate on cell number (animal size) is counterintuitive. The claim is made that large animals lose mass (volume) very slowly, while at the same time small animals display fast degrowth. If we understand the premise put forward, this would mean that small animals should disappear immediately given the high rate of degrowth.

Indeed, our demonstration that small animals degrow faster than large animals was unexpected. However, the inference of immediate disappearance of small animals is simply a misunderstanding of the data.Note that the growth rates reflect relative size change and that a negative growth rate simply reflects a decrease of size per unit time. E.g., a degrowth rate of 4%/day (as we measured for small worms) implies that it still takes about 17 days until a worm has degrown to half its initial size. Hence the size-dependence of the degrowth rate does not imply immediate disappearance. Why and if such counter-intuitive regulation of growth dynamics is adaptive in the wild is indeed an intriguing problem for further studies.

Moreover, it is also clear that the cellular content of the animal with a given size may affect the growth/degrowth rate. As such, it is difficult to follow the mathematical logic put forward to explain the recorded observations.

Indeed, changes in cell composition may impact the growth rate. However, our demonstration that the growth rate depends primarily on size identifies size as key rate determinant at the organismal level. If and how the organismal growth rate/size dependency reflects underlying size dependencies of cell types or cell composition constitutes a fascinating topic for future research, which we allude to in the Discussion section.

4) We have a few concerns regarding the energy balance models. We are including in a separate document (link below) a test of the mathematical models/assumptions. These were tested because we are having difficulty understanding the authors' proposal that the entirety of animal growth rate depends on energy per cell. Assuming a uniform distribution of animal energy over all cells is not justified – there might be strong fluctuations in the energy of individual cells (cephalic ganglia, parenchymal cells, and other organs, for example). Please consult our tests of Paradigms 1 and 3. We would appreciate hearing from the authors to help us understand the noted discrepancies.Link to document: https://submit.elifesciences.org/elife_files/2018/05/28/00052911/00/52911_0_attach_24_16058.pdf

This is a misunderstanding- we do not assume a uniform distribution of metabolic energy stores. Consider the case of the vertebrate liver that stores large amounts of ingested sugars in the form of glycogen. Nevertheless, the stored energy is subsequently made available to other tissues and organs. As such, glycogen stores in the liver are part of the global energy budget of the animal. Our concept of “energy/cell” reflects exactly the same coarse graining of a global energy budget and therefore neither assumes nor requires a uniform distribution of energy stores.